# Rotational fishing enables biodiversity recovery and provides a model for oyster (*Ostrea edulis*) habitat restoration

**Naomi A. Kennon**[1]*, **Alexander Robertson-Jones**[1], **Sebastian Jemmett**[1], **Tristan Hugh-Jones**[2], **Michael C. Bell**[3], **William G. Sanderson**[1]

**1** Institute of Life and Earth Sciences, School of Energy, Geoscience, Infrastructure and Society, Heriot-Watt University, Edinburgh, United Kingdom, **2** Loch Ryan Oyster Fishery Company LTD, Stranraer, United Kingdom, **3** International Centre for Island Technology, School of Energy, Geoscience, Infrastructure and Society, Heriot-Watt University, Edinburgh, United Kingdom

* nak7@hw.ac.uk

**Data Availability Statement:** All relevant data are within the paper and its Supporting Information files.

## Abstract

Reefs formed by *Ostrea edulis*, the European native oyster, are among many biogenic habitats that have declined globally. European oyster habitats are now rare, and undisturbed examples have not been described. As more is understood of the ecosystem services provided by the reefs, oyster restoration efforts are on the rise, becoming a more prominent component of Europe's portfolio of marine conservation practices. It is therefore important to establish the relationship between the development of oyster reefs and their associated biotic community if the biodiversity benefits are to be accurately predicted and the progress of restoration projects assessed. The Loch Ryan oyster fishery in Southwest Scotland is the last of its type and uses a rotational harvest system where different areas are fished each year and then left for six years before they are fished again. This provided an opportunity to study the effect of oyster reef development and biodiversity gain at different stages of habitat recovery. In this study three treatments were surveyed for faunal biodiversity, oyster shell density and oyster shell percentage cover. Treatments were plots that had been harvested one year before, two years before, and six years before the study. The treatments were surveyed with SCUBA using a combination of video transects and photo quadrats. Oyster shell density, oyster shell percent cover and macrofaunal biodiversity differed significantly between treatments, with the highest values observed in the six-year treatment. Shell density was 8.5 times higher in the six-year treatment compared to the one-year treatment, whilst Shannon-Wiener's diversity was 60.5% higher, and Margalef's richness 68.8% higher. Shell density and percent cover had a significant positive relationship with macrofaunal biodiversity. This is probably due to the provision of increased structural complexity in the matrix of live and dead oyster shells. Projecting forward the trend of biodiversity increase in relation to time since disturbance indicates that full recovery would take approximately ten years in which time diversity (Shannon-Wiener) would probably have doubled. The findings from the present study indicate the probable biodiversity benefits of oyster habitat restoration and a cost-effective metric (shell density) to judge progress in restoration projects.

**Funding:** The project received funding from The Glenmorangie Company (https://www. glenmorangie.com/en-gb) and Scottish Enterprise (https://www.scottish-enterprise.com/) as part of the Dornoch Environmental Enhancement Project (WGS: D18R11697). The funders had no role in study design, data collection and analysis, decision to publish, or preparation of the manuscript.

**Competing interests:** The authors have declared that no competing interests exist.

## Introduction

Habitat restoration to enhance biodiversity is becoming an increasingly popular conservation practice around the world in both terrestrial and aquatic habitats [1–4]. Many of these habitats have been lost or degraded due to anthropogenic activities [5]. Effective restoration requires an ecological understanding of the targeted habitat, but in many cases, we lack the data needed to determine the desired condition and biodiversity characteristics for a restored habitat. A reference habitat (analogous in terms of community composition, niche fulfilment and provision of ecosystem services) [6] can provide a baseline against which restorative progress can be judged and expectations of biodiversity outcomes clarified (see [7]).

In the last decade, restoration of the European native oyster (*Ostrea edulis*) has become a common component of coastal conservation projects in Europe [1,8]. Due to the rarity of intact *O. edulis* reef habitats (e.g., [8–12]) there is a lack of data with regards to the biodiversity and structure of the faunal community associated with the reefs [2].

Biogenic reefs made by bivalves, produce physically complex habitats formed from the dense coverings of living and dead shell, creating an intricate matrix of microhabitats [12–14]. The vertical structure of biogenic shellfish reefs can also provide sheltered areas and hydrodynamic relief [10,12], whilst clump structure of the bivalves provides escape from predation for juveniles and a multifaceted substrate for epibiota, egg laying and spawning [13–15]. Higher densities of bivalve shellfish can result in higher biodiversity [10,13] however no biodiversity records have been derived from undisturbed, mature oyster habitats. Even when Möbius (1877) described oyster habitats more than 140 years ago, they were already intensively fished.

In addition to providing physical niches for other species, biogenic shellfish reefs perform other provisioning, supporting and regulating roles [2,16–19]. These include bio-deposition [16,20] and providing essential fishery habitat [14,15]. The importance of ecosystem services supported by bivalve reefs is gaining recognition by government entities and industry [2,21,22]. This is also reflected by the increased investment by companies and NGOs into restoration projects. For example, the Glenmorangie Whisky Company, in partnership with the Marine Conservation Society and Heriot-Watt University have established the Dornoch Environmental Enhancement Project [13,23]. This initiative aims to restore native oysters to the Dornoch Firth whilst enhancing biodiversity and water quality in the system: the oyster filter feeding is expected to mitigate organic waste from the distillery's discharge water.

Despite the importance and value of shellfish habitats, many remaining reefs are at risk due to the spread of invasive non-native species, pollution, climate change and habitat disruption from the use of bottom towed fishing gear. Bottom towed fishing gear, such as those targeting scampi and scallops, comprise upwards of 25% of global fisheries [24]. The mechanical action of the dredge or trawl dragging across the seabed damages and kills many benthic organisms and communities [25]. Biogenic shellfish reefs are particularly vulnerable due to the fragility of the bivalves and the long regeneration rate and sometimes complex life history [26]. Damage and removal of biogenic reef forming organisms reduces structural complexity and creates a homogenised environment and a significant reduction in biodiversity [24,26].

Bivalves probably formed the most common type of biogenic reef in European temperate waters and were normally composed of *Mytilus edulis* (blue mussel), *Modiolus modiolus* (horse mussel) or *Ostrea edulis* (European native oyster) [2,16,17,26]. Of these, *O. edulis* was perhaps the most significant. There is evidence of oysters being a food source since before Roman times in many archaeological sites where oyster shell remains have been found [14,26]. Today, *O. edulis* has declined by more than 85% over the last century and shellfish reefs are considered some of the most imperilled marine habitats in the world [27]. During the 1800's and early 1900's oyster stocks were systematically exhausted to the point of commercial and functional

extinction. More recently, other factors have added pressure to oyster populations including pollution from chemicals such as tributyltin from antifouling paints, sedimentation and invasive species [26].

Due to the increased understanding of how important and vulnerable biogenic shellfish reefs are, actions to protect these now rare habitats are on the rise. Marine Protected Areas (MPA) now seek to limit damaging activities, allowing the habitat to recover naturally, but this is not always possible, especially in habitats which have suffered from chronic disturbance. Restoration as a conservation practice aims to actively restore habitats to resemble former states [27]. This can happen naturally once anthropogenic pressures have been removed but often requires building a self-sufficient ecosystem [28,29] by reintroducing or supporting the ecosystem engineers [30–32].

The significance of restoration projects is now acknowledged at an international level with support from industry, non-governmental organisations and by governments [32–34]. The UN Decade on Ecosystem Restoration aims to bring attention to the need to protect and restore ecosystems around the world. By 2030 the program aims to achieve a strong global movement working towards a sustainable future by restoring degraded habitats to the benefit of all society. Comprised of UN members, governments, NGOs and restoration practitioners, the program recognises the current needs of stakeholders and the need to preserve ecosystems which support them [35].

Oyster restoration is becoming a priority in many coastal conservation initiatives [8,14,35,36]. Many locations around Europe which supported historical oyster fisheries and /or remnant populations are now being considered as possible locations for *O. edulis* restoration [1,37,38]. Owing to the catastrophic decline and modern-day rarity of *O. edulis* reef habitats, and the historical context within which these declines occurred, it is extremely difficult to re-construct the biodiversity characteristics of a fully restored reef [39]. By examining the biodiversity of oyster habitats recovering from fishing activity [40], a greater understanding of the faunal community composition associated with oyster reef habitats can be gained [2,10,31] and used to manage expectations of biodiversity outcomes and progress towards biodiversity goals.

## Study aims

In the context of a long-term, sustained fishery for *O. edulis* in Loch Ryan (SE Scotland), the present study set out to investigate the relationship between the density of live and dead *O. edulis* shell and the biodiversity of the conspicuous macrofaunal community. The study also aimed to investigate biodiversity recovery linked to the rotational nature of the fishery. The implications of this work extend from identifying likely expectations for biodiversity development during and after restoration activities to the management of fished biogenic habitats and the biodiversity they support.

Hypothesis 1: The macrofaunal biodiversity of an oyster reef community is affected by the density of live and dead oyster shells.

Hypothesis 2: Biodiversity recovery of the macrofaunal community associated with an oyster reef is influenced by the length of time since disturbance.

## Methodology

**Study site.** Loch Ryan is a sheltered sea loch on the Southwest coast of Scotland where the Loch Ryan Oyster Fishery Company Ltd operates off Leffnoll Point, in the east (Fig 1). The fishery has been run by the Wallace family since 1701 and the fishery vessel uses a single

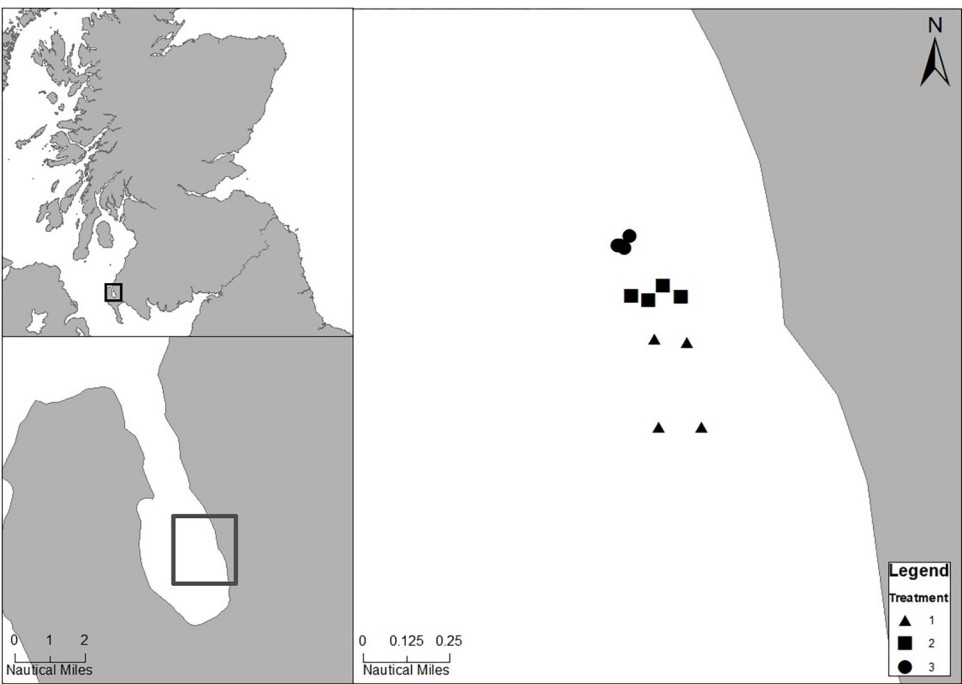

**Fig 1. Loch Ryan and the location of the study site off Leffnoll Point and sample locations.** (Map made using ArcGIS [Esri UK, Esri, 2022, Garmin, GeoTechnologies, Inc., USGS | Esri, 2022]).

dredge which only harvests the animals in the winter months, to avoid the spawning season. The oyster bed is divided into six plots which are harvested in a six-year rotation [40] and were regarded as treatments in the present study.

Three treatments were selected opportunistically for survey in 2019: Treatment 1 had been harvested one year before the study (2018), Treatment 2 had been harvested two years before (2017), and Treatment 3 had been harvested six years before (2013). Each treatment was surveyed in June 2019 using SCUBA. Four replicate transects were surveyed in each treatment (Fig 1). Sampling was conducted along a total of twelve 25m transects that ran parallel to the shore.

## Quadrat data

Each 25m transect was laid in the direction of tidal flow and eleven 0.5 x 0.5m quadrats were taken (S1 Fig). The quadrats were placed randomly, to avoid spatial auto-correlation, alternating between each side of the transect and photographed using a NIKON D200 DSLR camera in a Seacam underwater housing (Fig 2).

In each photo quadrat, the number of oyster shells was counted using the software ImageJ v1.52 by tracing around the shape of the individual shells [41]. These data were then averaged for each transect in each treatment.

## Video data

Video footage was taken of the seafloor covering a 0.5m swath on each side of the transect, representing a total of approximately six hours of footage. Difficult to identify macrofauna were recorded close-up for 5–10 seconds to assist in later identification. Experienced diving surveyors also recorded species *in situ*, collected voucher specimens, and these data were used for quality assurance of identifications made from video analysis (below).

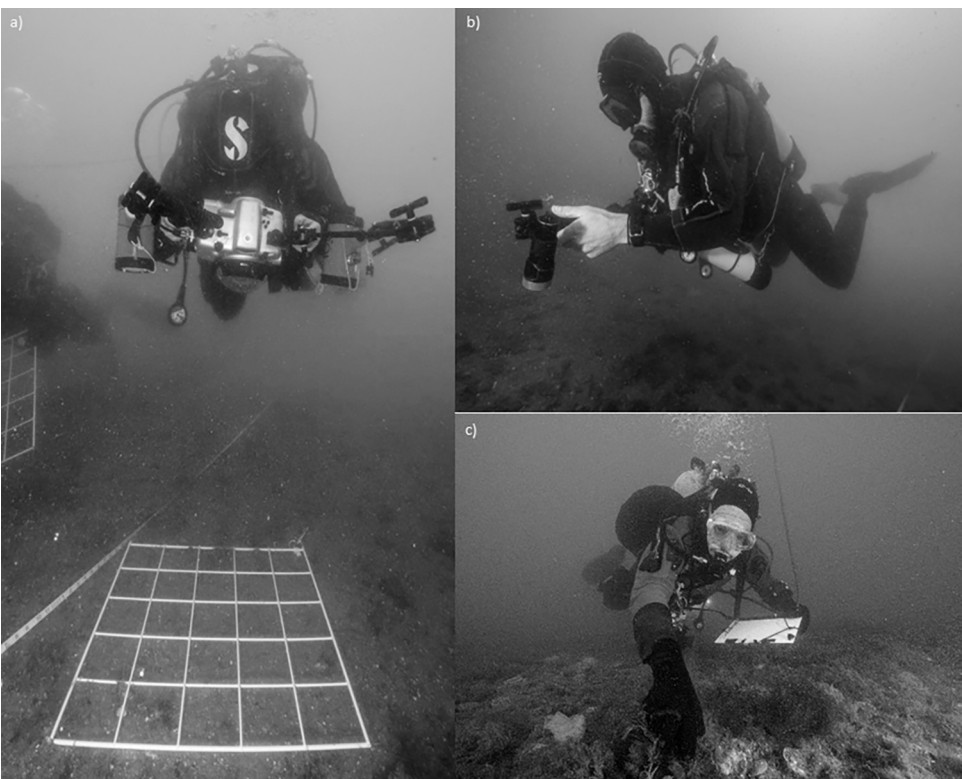

**Fig 2.** a) Scientific divers taking photo quadrats (NAK) and b) video recording transects (AJR). c) In situ recording (WGS). Photo credit: Richard Shucksmith.

The video footage was analysed to identify and count conspicuous macrofaunal species of demersal fish, crustaceans and echinoderms. Macrofauna species were identified to the lowest possible taxa given features visible in the video images. The video was played at half normal speed and screen grabs taken for quality assurance and species validation by the field team. To assist in identification and avoid learning bias in the data set, a training exercise was conducted before commencing the main analysis by using a selection of random transects to gain familiarity with the species present.

### Data analysis

Primer v7 [42] was used for analysis of species composition, including calculation of the biodiversity indices: Margalef's richness (d) a measure based on the number of species present and sampling effort; Pielou's evenness (J) a measure of how evenly species are distributed; and Shannon-Wiener's diversity (H') which takes both richness and evenness into account. To down-weight the influence of highly abundant taxa (1 order of magnitude in variation), species abundance data were first square-root transformed before being used to calculate a Bray-Curtis similarity matrix which was then used to compare biodiversity, oyster shell measures and benthic features in a non-metric Multidimensional Scaling (nMDS) plot. SIMPER and ANOSIM analyses were used to compare species composition between the macrofaunal communities of each treatment based on Bray-Curtis similarity matrices.

Univariate statistical analyses were undertaken in Minitab v19 [43]. ANOVA was used to compare biodiversity indices between treatments and Tukey pairwise comparisons as a post-hoc test to compare between groups. Regression models using logarithmic transformation

were created to compare and predict the effect of oyster density and oyster shell cover on macrofaunal biodiversity. A logarithmic model was selected on the basis of explanatory power and approximating to the asymptotic nature of the recovery process (diminishing rate of increase as reef approaches 'recovered' condition). The response variable was modelled as a function of the log of years since disturbance to represent this non-linear (logarithmic) process.

## Results

### Biodiversity and shell density

A total of 27 macrofaunal species were recorded in the course of the surveys (S1 Table).

### Community change

The nMDS plot (Fig 3) illustrates a clear trend of community change from left to right over axis 1. The stress value (2D stress = 0.02) is extremely small, indicating that this plot is an accurate representation of the overall pattern of dissimilarity between samples.

The macrofaunal community in Treatment 1 was significantly different from Treatment 3 (ANOSIM, r = 0.51, p = 0.029) but Treatment 2 did not differ significantly from the others. SIMPER analysis confirmed the highest dissimilarity was between Treatments 1 and 3 (65.5%) and the least dissimilarity between Treatment 2 and 3 (42.5%). Ten species contributed more than three-quarters of the total dissimilarity between treatments 1 and 2, and 40% was attributable to increases in the abundance of *Pagurus bernhardus* (Linnaeus 1758), *Antedon bifida* (Pennant 1777) and *Hyas araneus* (Linnaeus 1758) (Table 1).

### Statistical models

Compared with the samples taken from the treatment one year after fishing, Shannon-Wiener's diversity was 60.5% higher in samples taken six years after fishing, Margalef's richness was 68.8%

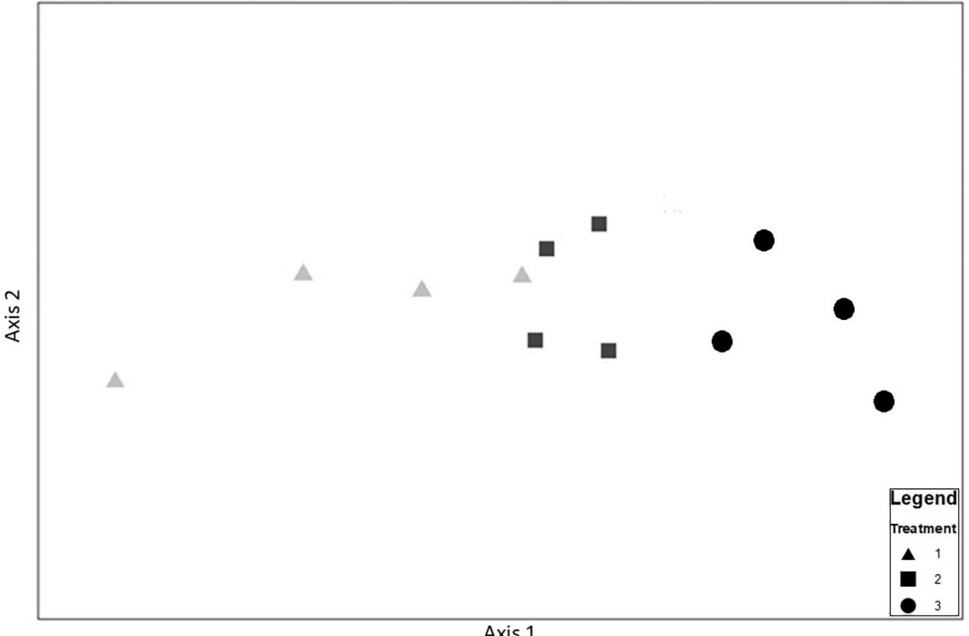

**Fig 3. Changes in benthic macrofaunal community with years since disturbance (2D stress = 0.02).**

**Table 1. SIMPER analysis comparing percentage dissimilarity between Treatment site 1 and Treatment site 3.**

| | Treatment 1 | Treatment 3 | | |
|---|---|---|---|---|
| Species | Average Abundance | Average Abundance | Contribution % | Cumulative % |
| Pagurus bernhardus | 1.76 | 4.**61** | 17.21 | 17.21 |
| Antedon bifida | 0.83 | 2.84 | 13.52 | 30.72 |
| Hyas araneus | 0.25 | 2.05 | 9.27 | 39.99 |
| Ophiothrix fragilis | 0 | 1.62 | 7.71 | 47.7 |
| Inachus dorsettensis | 0.35 | 1.21 | 5.2 | 52.91 |
| Gobius niger | 0.93 | 1.41 | 5.08 | 57.99 |
| Carcinus maenas | 0.5 | 0.99 | 4.92 | 62.91 |
| Crossaster papposus | 0 | 0.93 | 4.9 | 67.81 |
| Pomatoschistus microps | 1.92 | 2.15 | 4.81 | 72.62 |
| Liocarcinus depurator | 0.85 | 1.49 | 3.42 | 76.04 |

higher whilst Pielou's evenness was 4.3% lower. Correspondingly, average oyster shell density increased between treatments 1 and 2 by 749.5%, i.e. a more than eight-fold increase.

Logarithmic regression (Fig 4) showed a strong trend between shell density and years since disturbance.

Likewise, a similar trend was shown when comparing biodiversity (Shannon-Wiener's diversity index) to time since disturbance (Fig 5).

Logarithmic models demonstrated the relationship of macrofaunal biodiversity with average oyster shell density (Fig 6). Shannon-Wiener's diversity (6a) and Margalef's richness (6b) were found to be positively correlated to both shell density and shell cover however, oyster shell density showed the strongest relationships (Fig 6) and no relationship was apparent for Pielou's evenness (6c) in either case.

## Discussion

The present study showed that oyster shell had a positive influence on macrofaunal diversity. Furthermore, the accumulation of shell and associated increase in biodiversity is likely a

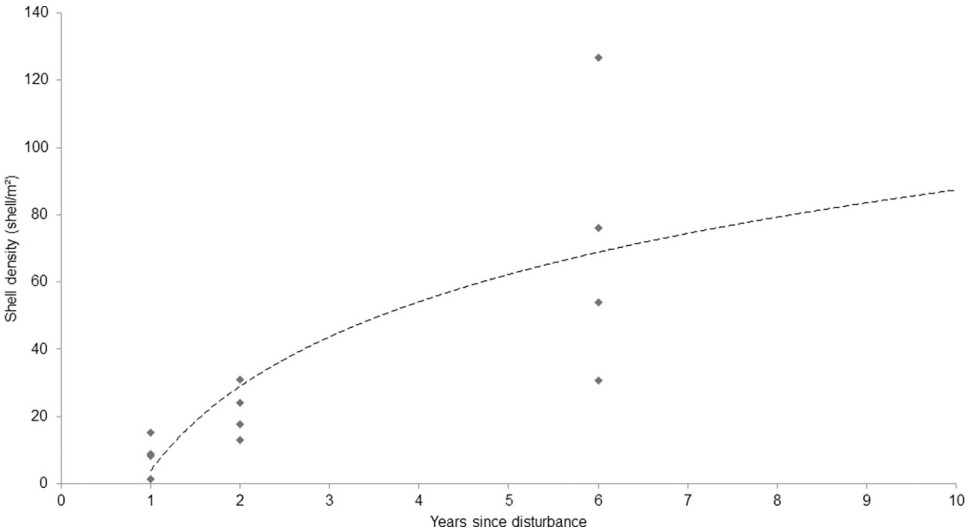

**Fig 4. Logarithmic model of shell density by years since disturbance.** $y = 36.327\ln(x) + 3.785$ $R^2 = 0.6023$.

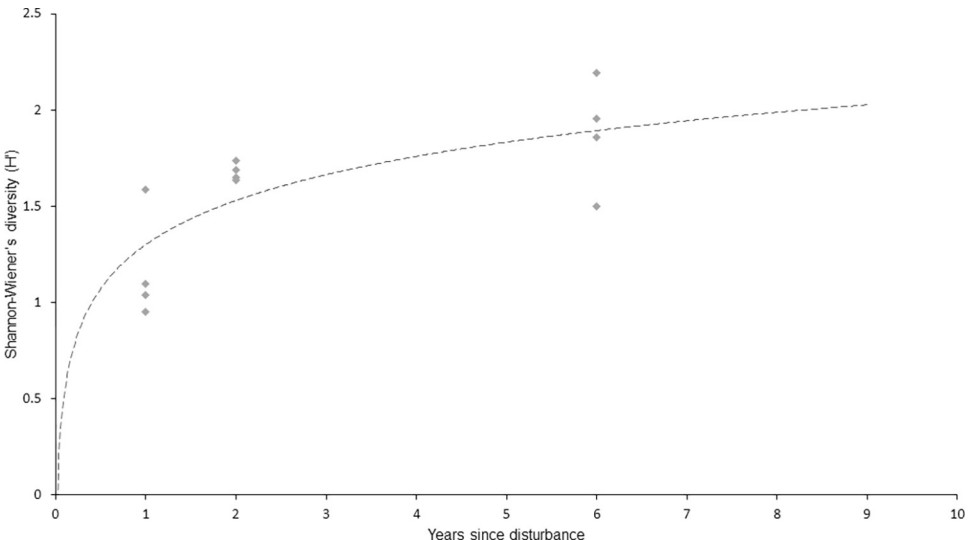

**Fig 5. Logarithmic model of biodiversity by years since disturbance.** $y = 0.3752\ln(x) + 1.2635$ $R^2 = 0.5881$.

function of time since fishing with a recovery trajectory based on logarithmic models (Fig 5.) of approximately ten years. The two hypotheses are therefore supported; the macrofaunal biodiversity of an oyster reef community is affected by both the density of oyster shells and the time since disturbance. This has important implications for the management of biodiversity and the expectations of oyster habitat restoration by managers and practitioners.

In keeping with most biodiversity studies [44,45], the sampling method presented here is expected to represent community-wide patterns. Potential confounding environmental factors were not evaluated (eg depth, temperature and tidal flow rate) because they were judged to be diminutive. Fishing is the dominant structural influence on the community, hence the results from the study are unlikely to be the result of environmental gradients that could produce

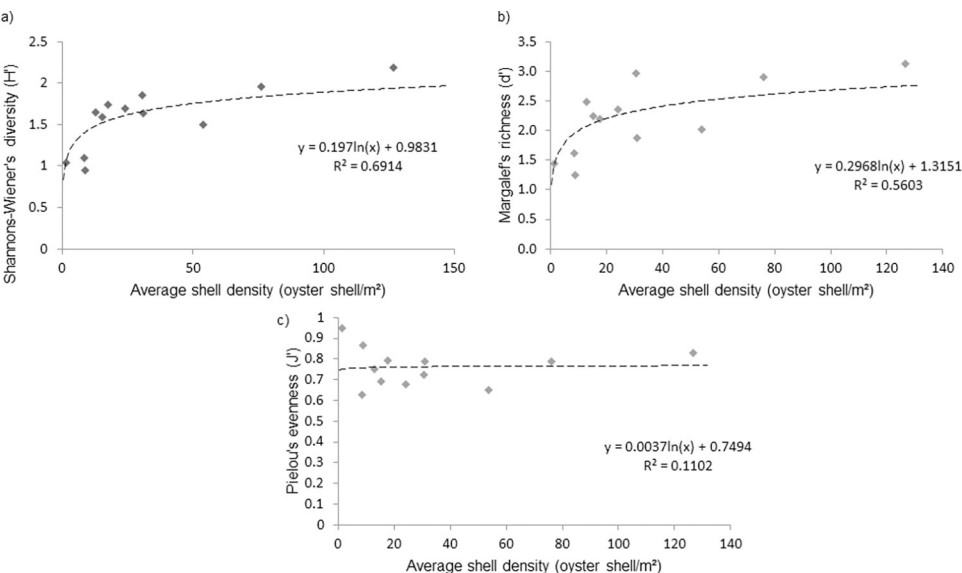

**Fig 6.** Regression models comparing biodiversity indices against shell density a) Shannon-Wiener diversity b) Margalef's richness c) Pielou's evenness.

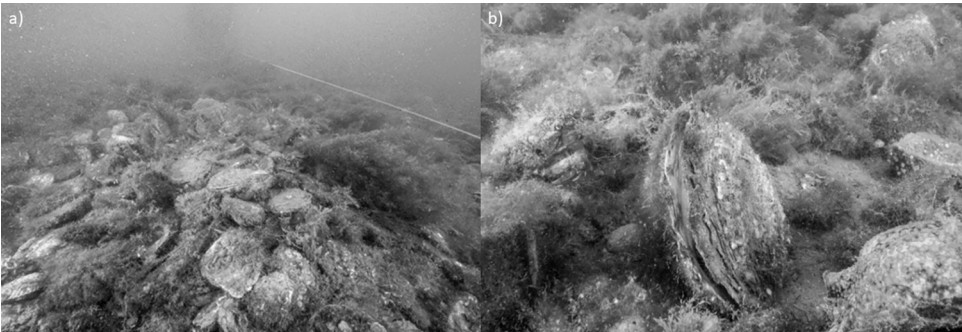

**Fig 7. Examples of a developed oyster reef in Loch Ryan Treatment 3.** a) An area of oyster reef demonstrating high density of adult sized oysters. b) A close up of an adult sized oyster. (Photo credit: Richard Shucksmith).

such a significant scale of change. Furthermore, the scale and trend in the observed relationships are consistent with successional differences [10,25,46,47] not spatial patterns.

The biodiversity trends observed in the macrofaunal community across the three treatments are consistent with the six-year rotational harvesting system operated by the Loch Ryan Oyster Fishery Company Ltd [40,48]. Elsewhere, bottom towed fishing gear has been shown to substantially reduce the diversity and productivity of biogenic reefs and their associated faunal communities [25,49] and lead to the destruction of extensive shellfish habitats [46,50]. In the present study, however, oysters appear to have been allowed time to resettle and grow after each fishing season, allowing macrofaunal biodiversity to recover by 60% after six years and re-establish a diverse macrofaunal community. Ordinarily, bottom-towed fishing gear has been widely implicated in the destruction of the biodiversity of physically complex habitats around the world [25,46,51,52]. Indeed, oyster habitats were extirpated throughout Europe and the world using unsustainable bottom-towed fishing practices in the 1800's and early 1900's [9,25,26,53,54]. The rotational system of management in Loch Ryan, however, is unusual for a dredge fishery [48], apparently allowing sufficient reef recovery to maintain biodiversity hotspots in the loch (Figs 7 and 8).

The recovery of bivalves is dependent on successful larval settlement, recruitment, and growth, suggesting the need to also maintain overall adult stock levels as a source in any restoration or sustainable management scenario. Reef recovery likely begins as early as one year after fishing disturbance and increases over several more years as oyster spat settles and the size, density and complexity of oyster shell increases (Figs 7 and 8) [3,26,47]. Beyond the six-year timeline studied here the biodiversity trajectory is expected to plateau out, but examination of oysters in an undisturbed state would be required to confirm this. Elsewhere in longer term (10yr+) marine reserves studies, 'trophic cascades' have occurred when re-established

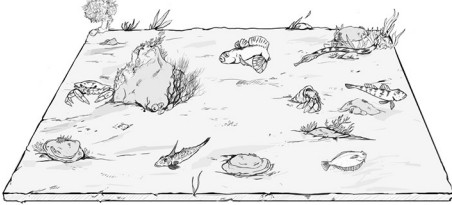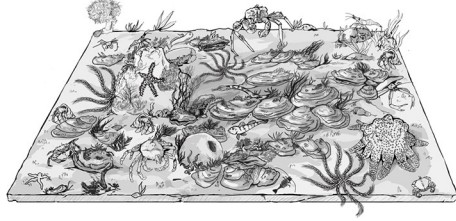

**Fig 8. Infographic demonstrating the change in biodiversity over six years.** (Credit: Siôn Williams).

higher trophic level species have restructured benthic communities beyond simple biodiversity and biomass increases that were initially observed [eg. 55]. A six year timeline in the present study is too short to exclude the possibility that trophic cascades ultimately transform the benthic community and affect oyster reef biodiversity.

Logarithmic regression showed positively correlated relationships between biodiversity, oyster shell density and time since disturbance. Oyster shell density was the best statistical predictor of Shannon-Wiener's diversity, which is likely to be due to the positive relationship between shell density and structural complexity [11,13]. Lown et al. (2021) demonstrated a positive relationship between oyster density and biodiversity using dredge sampling in a remnant oyster fishery in England [11]. The present findings are consistent with Lown et al. (2021), confirming a strong relationship between biodiversity and shell density [11]. This is likely due to the influence of structural diversity created by the shell material. Predicting biodiversity from shell density could prove a useful, cost-effective indicator of the condition of a fished or restored oyster habitat.

Three indices measuring different components of diversity were tested in this study, of which, Shannon-Wiener's diversity and Margalef's richness were found to vary significantly. Pielou's evenness however, was found not to change significantly in relation to time since fishing. This indicates that as the oyster reef becomes more developed, the number of species increases, but the relative abundances between species within the reef community does not significantly change.

The abundance of three species which made up a large proportion of the difference between the species composition of Treatment 1 and Treatment 3 (Table 1) were *Pagurus bernhardus* (Linnaeus 1758), *Antedon bifida* (Pennant 1777) and *Hyas araneus* (Linnaeus 1758). This difference can likely be explained by the higher number of oyster shells found in Treatment 3. It is probable that the oyster shells interact with each other to provide physical structure with high surface area to volume ratio and therefore increased structural complexity [15,36]. *A. bifida* is a fragile species which is particularly susceptible to disturbances such as dredging, however greater structural complexity provides attachment points higher in the water column allowing for more efficient filter feeding [15,45]. *P. bernhardus* and *H. araneus*, as mobile taxa, are less affected by disruption caused by dredging than other sessile taxa [49]; however, greater structural complexity provides more places to shelter as well as greater feeding opportunities [15].

In the last five years there has been a rapid increase in the number of restoration projects targeting *O. edulis* habitats, many of which aim to improve biodiversity [1,8] but very few, if any, have demonstrated this [2]. The relationship between oyster shell and biodiversity presented here is perhaps the most direct evidence yet quantifying the potential biodiversity gain (60% or more over six years) that can be achieved as oyster habitats develop (Fig 8). Furthermore, the projection of Shannon diversity (Fig 5) indicates that biodiversity would likely double in a decade.

Overall, the present findings demonstrate the benefits of restoration and the trajectory of restoration success in terms of biodiversity [1,35,36]. Our data showed a link between increased shell material and increased biodiversity. The present study is a rare example from a long-term dredge fishery [40,48] whose practices appear to have allowed rare oyster habitat and the associated community to persist, thereby providing a valuable insight into the recovery of biodiversity by balancing sustainable fishery practices with European oyster habitat recovery efforts.

## Supporting information

**S1 Fig. Transect survey layout.**
(TIF)

**S1 Table. Species abundance data from treatment 1, 2 & 3 & S3 Table.** Average oyster shell density and percentage cover data for treatments 1,2 & 3.
(DOCX)

## Acknowledgments

This research supports the Dornoch Environmental Enhancement Project: A project that seeks to restore native oysters to the Dornoch Firth. We are very grateful to Heriot-Watt Scientific Divers, especially Hamish Mair, Dan Harries, Owen Paisley, Hannah Lee and Stephanie Lapidge for additional field support, as well as Rab Lamont and John Mills of Vital Spark. We would also like to thank Richard Shucksmith for the use of his imagery and Siôn Williams for creating the biodiversity infographic. Comments made by the reviewers have greatly improved the manuscript.

## Author Contributions

**Conceptualization:** Naomi A. Kennon, William G. Sanderson.

**Data curation:** Naomi A. Kennon.

**Formal analysis:** Naomi A. Kennon.

**Funding acquisition:** William G. Sanderson.

**Investigation:** Naomi A. Kennon, Alexander Robertson-Jones, Sebastian Jemmett, William G. Sanderson.

**Methodology:** Naomi A. Kennon, William G. Sanderson.

**Project administration:** Naomi A. Kennon, William G. Sanderson.

**Resources:** Tristan Hugh-Jones.

**Supervision:** Michael C. Bell, William G. Sanderson.

**Visualization:** Naomi A. Kennon.

**Writing – original draft:** Naomi A. Kennon.

**Writing – review & editing:** Michael C. Bell, William G. Sanderson.

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
