## [Decision Letter · Decision Letter 0]

22 Jul 2022

PONE-D-22-14818Rotational fishing enables biodiversity recovery and provides a model for oyster (*Ostrea edulis*) habitat restoration PLOS ONE

Dear Dr. Kennon,

Thank you for submitting your manuscript to PLOS ONE. After careful consideration, we feel that it has merit but does not fully meet PLOS ONE’s publication criteria as it currently stands. Therefore, we invite you to submit a revised version of the manuscript that addresses the points raised during the review process.

We look forward to receiving your revised manuscript.

Kind regards,

José A. Fernández Robledo, Ph.D.

Academic Editor

PLOS ONE

Journal Requirements:

    "This research was supported by The Glenmorangie Company and Scottish Enterprise as part of the Dornoch Environmental Enhancement Project: A project that seeks to restore native oysters to the Dornoch Firth. We are very grateful to Heriot-Watt Scientific Divers, especially Hamish Mair, Dan Harries, Owen Paisley, Hannah Lee and Stephanie Lapidge for additional field support, as well as Rab Lamont and John Mills of Vital Spark. We would also like to thank Richard Shucksmith for the use of his imagery and Siôn Williams for creating the biodiversity infographic.

   "The project received funding from The Glenmorangie Company (https://www.glenmorangie.com/en-gb) and Scottish Enterprise (https://www.scottish-enterprise.com/) as part of the Dornoch Environmental Enhancement Project (WGS: D18R11697). 

3. Please ensure that you include a title page within your main document. You should list all authors and all affiliations as per our author instructions and clearly indicate the corresponding author.

5. We note that Figure 2 includes an image of a participant in the study. 

6. Please ensure that you refer to Figure 3 in your text as, if accepted, production will need this reference to link the reader to the figure.

Reviewers' comments:

Reviewer's Responses to Questions

**Comments to the Author**

1. Is the manuscript technically sound, and do the data support the conclusions?

Reviewer #1: Yes

Reviewer #2: Partly

2. Has the statistical analysis been performed appropriately and rigorously? 

Reviewer #1: Yes

Reviewer #2: Yes

3. Have the authors made all data underlying the findings in their manuscript fully available?

Reviewer #1: No

Reviewer #2: Yes

4. Is the manuscript presented in an intelligible fashion and written in standard English?

Reviewer #1: Yes

Reviewer #2: Yes

5. Review Comments to the Author

Reviewer #1: The approach of using a rotational fishery to study reef recovery and biodiversity is insightful, and this study will serve as a useful resources for restoration practitioners. I have a number of comments below, which would help with the clarity have some sections of the manuscript.

Lines 59-61 - Would benefit from references about the risks to shellfish reefs. Also surprised that climate change (and synergistic effects) not listed among the risks in the long-term.

The manuscript would benefit from a clearer description of the methods used in a few areas. Specifically:

Lines 115-117 (study site) - Suggest adding more site description. For instance - What are the depths of the reefs being surveyed? Other than a rotational system, how does the Loch Ryan Oyster Fishery Company operate (e.g., gear type, season, etc.)?

Lines 120-121 - The four replicate transects in each plot are arranged quite differently (treatment 1 – in a rectangle, treatment 2 – in a line spread out, treatment 3 – clustered together). How were the locations of the four transects in each plot selected, and what if any impact might that have on the results? Do the characteristics of each transect vary at all?

Lines 123-125 - Clearer description of the photo quadrat layout. The picture of the quadrats in figure 2 depicts a grid of 25 quadrats. Were a random sample of those selected at each location along the transect, or just one .5x.5m quadrat photographed/counted at each location, randomly selected along the 25m transect?

Figure 3 is missing axis labels/titles, making it hard to interpret

Figure 4 - In the plot with 6 years since disturbance, why might the shell density vary so much between the four transects? More generally, can the authors provide any more description of the variability within each treatment, across different metrics?

Lines 189-190 - What is the relationship between shell cover and diversity/richness? Reported as strongly related but the correlation/significance is not specified.

Line 198 - Where did the recovery trajectory of ten years come from? Is this from the literature (if so, needs citation) or the study?

Line 213-215 - Minor clarification, would suggest adding "after 6 years" after "allowing macrofauna biodiversity to recover by 60%". As currently written, could be misinterpreted to mean that biodiversity recovers by 60% after each fishing season.

Data availability - Data on percent oyster shell cover were not available, but the remaining data were.

Reviewer #2: Line 22: Change to “percent cover”--also, is this percent cover of benthos that is shells? To make clearer, the authors could consider changing this metric to ‘benthic shell cover’?

Line 23: Since there are multiple plots within each treatment, I would replace the word “plot” with treatment. If the authors made this edit they could then rephrase the following sentence to “Shell density was 8.5 times higher in the six-year treatment compared to the one-year treatment, whilst…”.

Line 26: Change “percentage” to “percent”

Line 31: Rephrase slightly, perhaps by starting with the result and deleting the first section of the sentence and start with something like-- “Here we show that the likely scale of biodiversity gain….”. The authors end with how this can be used for conservation and management so the first half of the sentence is unnecessary.

Line 35: I suggest the authors open with a broader introduction to restoration and the need for reference sites, data, etc. This would provide more context for the first paragraph which would follow and would make the paper applicable to a broader audience. This is a huge issue/topic of interest in the field of restoration and could be fleshed out more. The authors could then use their study to help provide that reference data that is so critical for evaluating project outcomes.

Line 35: Define reference model. “Reference” is typically reserved for a natural, un-impacted site but no reference was mentioned in the abstract so from this point on the reader could be thinking that the 6-year treatment is the reference. If there is a reference site included, that should be mentioned in the abstract as the “target”.

Line 40: What does “familiar” mean here? I would rephrase. Who is the audience for this paper? If managers, as was set up in the abstract, these individuals would know of rocky reefs biogenic reefs. Also, since most bivalves require some hard substrate upon which to settle, maybe rephrase to emphasize instead that reef structural complexity can be constructed from bedrock, but (often overlooked) bivalves too can provide (more?) intricate microhabitats.

Lines 46-49: Strongly encourage the authors to integrate natural, undisturbed reefs into the abstract given that this has been the focus thus far of the introduction.

Line 50: Replace “As well as” with “In addition to providing physical…”

Line 52: Need comma after psuedofaeces

Line 54: Remove “an”

Lines 56-58: Rephrase to make more concise. Unless introducing monetary value of the services, I’d rephrase to “importance”. Something like “The importance of ecosystem services supported by bivalve-built reefs is gaining recognition by government entities and industry.” It would be great if the authors could point to an example of this increased recognition, there are certainly numerous examples. Perhaps the authors could cite a few papers that lay out examples that support the sentence. When editing this sentence, I recommend the authors remove the last portion of the sentence “for the benefits…” since that is implied when mentioning ecosystem services in the beginning of the sentence.

Line 76: should read “including” not “included”

Line 79: Rephrase to something like “Coincidence with the increased understanding of how important and vulnerable biogenic shellfish reefs are, actions to protect this now rare habitat are now also on the rise.”

Line 81: I would clarify that oyster restoration is a relatively new approach--in terrestrial and (less so) marine habitats, it is not a new approach. If the authors choose to keep this sentence as is, I suggest clarifying the conservation/management strategy they are comparing restoration to that validates restoration as “relatively new”. I encourage the authors to re-write this sentence since I do not think it is accurate, as written.

Line 83: Sometimes restoration plans include plans for chronic intervention (if necessary) so I would add “often” to the sentence--“This often requires building a self-sufficient…”. Additionally, a sentence on passive restoration to encourage natural recovery could be included here.

Line 85: Topic sentence is very similar to previous sentences above. I would rephrase.

Line 93: Remove first clause of sentence, start paragraph with “Oyster restoration is…”

Line 98: Replace “articulate” with “re-construct” and revise sentence--“…it is extremely difficult to re-construct the biodiversity characteristics of a fully restored reef, given how rare pristine habitats have become”

Lines 99-101: Rephrase to better transition from previous sentence. It’s my understanding that the authors are trying to state that studies that compare biodiversity in restored and less disturbed/somewhat natural habitats is necessary for measuring progress towards reaching biodiversity goals. I would re-write this last sentence to reflect this study goal, that will also lead into study aims nicely.

Line 106: Rewrite slightly--Something like…“The implications of this work extend from identifying likely expectations for biodiversity development post-restoration to the management of fished biogenic habitats and the biodiversity they support.”

Line 111: Could rephrase Hypothesis 2 to clarify language -- “Recovery of macrofaunal community biodiversity is influenced by time since disturbance”

Hypothesis Question: The authors have not defined biodiversity yet and this term is often overused or misused. It may be nice to define this in the introduction so the reader is aware ahead of time whether you are talking about biodiversity as species richness and evenness (using H’) or some other variable like species density, which is being used as a proxy for biodiversity.

Since biodiversity is a key part of this study, the authors could add more to the introduction and discussion as to why biodiversity is ecologically important. Provide specific examples, cite relevant literature, etc. The majority of the intro was focused on ecosystem services and the importance of biogenic habitats but the ecological importance of the diverse assemblage of species that these habitats support could be strengthened and made more prominent in the intro.

Line 118: I’d remove the term “plots” and only refer to them as Treatment 1, 2, and 3, each with four replicate transects. Ideally, the authors would have tracked each Treatment as it evolved since there may be differences between treatments unrelated to treatment (i.e., distance to shore, hydrology, relief of bedroom, depth, turbidity, etc.).

Authors should address the above (and other) potential confounding factors. If these factors were not evaluated during the study period, their omission should be acknowledged as a potential explanation that could weaken the strength of the findings. Looking at Figure 1 the Treatments are not interspersed and are clumped, so the potential confoundment of the results due purely to environmental factors and not treatment needs to be addressed (if the data was not collected and shown to be similar across treatments). If this environmental data exists, it should be highlighted in the section that outlines the details for all the various treatments.

Is there no reference habitat? Is Treatment 3 serving as the reference habitat? Since this was a prominent topic in the introduction, I suggest aligning the experimental design to the motivation outlined in the intro.

Line 124: Why were quadrats placed randomly instead of at some standardized interval? Could the authors elaborate on this? Were they opportunistically sampling along the transect line on whatever side had the most biogenic habitat? I suggest the authors justify the sampling design.

Line 135: “were taken for…”

Figure 2 Caption: “Scientific divers taking photo…”

Line 146: Add Figure 3 for NMDS? Missing caption from this section?

Figure 3: Add legend. Match symbols between Fig 1 and 3

Line 142-143: I’ll defer to the other reviewers here to verify, but I don’t believe Pielou’s evenness should be run on transformed data. If the data were transformed for evaluating evenness, what was the justification?

Line 179: Evenness (add “n”)

Line 197: Replace “probably” with “likely”

Line 198: How are the authors defining recovery trajectory? The curve plateauing? What if the plateau in the fished areas is still below what would have been the biodiversity “target” of a natural undisturbed oyster reef? This has not been addressed yet but is critical.

If ten years is the “recovered” state, Fig 4 should extend to 10 like Fig 5 does since shell density, which covaries with time since harvesting, is stated as the ultimate cause of increased biodiversity.

Line 199: Replace “accepted” with “supported”.

Line 200: The authors have yet to acknowledge that what they may be witnessing is a version of the intermediate disturbance hypothesis. What if natural reefs have lower biodiversity? It’s important to acknowledge that the 10-year “recovery” is based on a disturbed system. The authors should include text that acknowledges 1) no reference included, 2) confounding factors could explain differences in biodiversity, 3) have not tracked individual treatments over time, only a snapshot in time so can’t rule out #2.

Line 206: I don’t think this is an issue at all actually since the level of detail in sampling was the same across all treatments.

Line 224: Why inevitably?

Line 225: What is several years? 2-3 or 5-6? Please specify.

Lines 223-225: Is this where the ten years comes from?

Lines 223-225: The study results show that recovery begins as early as 1 year post-harvest and increases over time as shell density increases, but without knowing what biodiversity would be without any harvest, it’s unclear that the recovery would be on a ten year cycle when the fished system is on a 6-year cycle. Oysters would be recruiting throughout that 6-year period and the contributions of each cohort to overall shell density would be staggered. It’s also possible there is a benefit to biodiversity when the size structure of the oyster reef is varied and that’s what’s driving the pattern (another confounding factor that should be mentioned in the discussion). Treatment 3 is going to have a broader size frequency distribution than Treatment 2 or 1. Since this data was not collected, the contributions of a reef composed of different sized individuals should be addressed.

Figure comment: Have symbols match across all figures, when applicable.

Line 233: remove “aspects of the”

Line 236: This sentence about the indices of different components of diversity should be included in the introduction under study aims as well in the discussion (see above comment on this)

Lines 237-240: Could this be due to the down-weighting of abundances when transformed? Please clarify whether the data used for the evenness index was transformed. If it was transformed, justify. If it was not transformed, include text that distinguished what data was vs was not transformed.

Line 242: “recently fished plots (Treatment 1) and those that had been fished six years previously (Treatment 3)”

Line 244: “less recently fished plot” replace with Treatment 3

Line 255: The biodiversity gain is relative to Treatment 1, correct?

Line 257: Figure 8 should not be cited here since this is an infographic showing change in six years. I recommend moving the reference to Fig 8 to Line 256 after “that can be achieved as oyster habitats develop”.

Line 260: Replace “simultaneous” with “concurrent”

Line 260: Remove “per se”

Line 261: Is 1701 the age of the Loch Ryan Company? If so, that detail should be in Intro or Methods.

Discussion Comment: I strongly encourage the authors to address the limits of the data they collected. Without environmental data, their findings could be confounded by factors outlined in previous comments above. Additionally, the authors did not track each individual Treatment over time so there is no way to know whether Treatment 1 in 2025 would look like Treatment 3 did in 2019. This assumption, given that their locations do not overlap, needs to be laid out in the main text.

Overall, the paper was well-written and easy to follow. The findings were compelling but the limitations of the study, given the limited data collected, need to be acknowledged in the main text.

It would be great if the authors could apply their findings and more strongly recommend management options for oyster aquaculture and more fully flesh out the benefits of allowing a biodiverse community to develop both ecologically and economically (for the fishery).

Great work! With these revisions, this paper will be a valuable contribution to the field.

-Kat Beheshti

6. PLOS authors have the option to publish the peer review history of their article (what does this mean?). If published, this will include your full peer review and any attached files.

Reviewer #1: No

Reviewer #2: **Yes: **Kathryn Beheshti

---

## [Author Response · Author response to Decision Letter 0]

5 Sep 2022

Dear José A. Fernández Robledo, Ph.D. 05/09/22 

Thank you for giving me the opportunity to submit a revised draft of my manuscript titled

Rotational fishing enables biodiversity recovery and provides a model for oyster (Ostrea edulis) habitat restoration to Plos One. We appreciate the time and effort that you and the reviewers have dedicated to providing your valuable feedback on my manuscript. We are grateful to the reviewers for their insightful comments on my paper. We have been able to incorporate changes to reflect most of the suggestions provided by the reviewers. We have highlighted the changes within the manuscript.

See the following table for a point-by-point response to the reviewers’ comments and concerns.

Line Comment Response New line

Acknowledgements Funders should not be listed in the acknowledgement Removed funders from acknowledgements 

Title page Please ensure that you include a title page within your main document. You should list all authors and all affiliations as per our author instructions and clearly indicate the corresponding author. Corrected 

Figure 1 Copyright Map was created using ArcGIS using shapefiles from ESRI.

Esri UK, Esri, 2022, Garmin, GeoTechnologies, Inc., USGS | Esri, 2022]) 

Figure 2 Consent for photo Photo is of Scientific Divers conducting survey. Not of study subjects. 

Figure 3 Refer to figure in text Corrected 219

Supporting files Include captions for supporting information files at the end of the manuscript http://journals.plos.org/plosone/s/supporting-information Corrected 

22 Change to “percent cover”--also, is this percent cover of benthos that is shells? To make clearer, the authors could consider changing this metric to ‘benthic shell cover’? Changed to “Oyster shell density, oyster shell percentage cover” as specific to oyster shell cover 27

23 Since there are multiple plots within each treatment, I would replace the word “plot” with treatment. If the authors made this edit they could then rephrase the following sentence to “Shell density was 8.5 times higher in the six-year treatment compared to the one-year treatment, whilst…”. Corrected 

31 Rephrase slightly, perhaps by starting with the result and deleting the first section of the sentence and start with something like-- “Here we show that the likely scale of biodiversity gain….”. The authors end with how this can be used for conservation and management so the first half of the sentence is unnecessary Changed to “The findings from the present study indicate the likely scale of biodiversity gain in oyster habitat restoration and a cost-effective tool (shell density) to judge progress in restoration projects.”

 36

35 I suggest the authors open with a broader introduction to restoration and the need for reference sites, data, etc. This would provide more context for the first paragraph which would follow and would make the paper applicable to a broader audience. This is a huge issue/topic of interest in the field of restoration and could be fleshed out more. The authors could then use their study to help provide that reference data that is so critical for evaluating project outcomes. Changed to:

“Habitat and biodiversity restoration is becoming an increasingly popular conservation practice around the world in both terrestrial and aquatic habitats [1][2][4][5]. Many of these habitats have been lost or degraded due to anthropogenic activities [3], making them rare or endangered. For restoration to work effectively there must be an ecological understanding of the original habitat, but in many cases, due to the degradation or rarity of the habitat, such studies are not available. A reference habitat (analogous in terms of community composition, niche fulfilment and provision of ecosystem services)[6] can provide a baseline against which restorative progress can be judged and expectations of biodiversity outcomes clarified (see [7]).”

 40

35 Define reference model. “Reference” is typically reserved for a natural, un-impacted site but no reference was mentioned in the abstract so from this point on the reader could be thinking that the 6-year treatment is the reference. If there is a reference site included, that should be mentioned in the abstract as the “target”. Create “A restoration practice that has become common in coastal conservation projects in Europe is that of the native oyster (Ostrea edulis) [1][8]. Due to the rarity of intact O. edulis reef habitats (e.g., [8][9][10][11]) there is a lack of knowledge of their composition and biodiversity [2]. It has therefore been necessary to refer to analogous habitats to fill knowledge gaps and build an understanding of a restored oyster reef.” 48

40 What does “familiar” mean here? I would rephrase. Who is the audience for this paper? If managers, as was set up in the abstract, these individuals would know of rocky reefs biogenic reefs. Also, since most bivalves require some hard substrate upon which to settle, maybe rephrase to emphasize instead that reef structural complexity can be constructed from bedrock, but (often overlooked) bivalves too can provide (more?) intricate microhabitats. Deleted this line. 

46-49 Strongly encourage the authors to integrate natural, undisturbed reefs into the abstract given that this has been the focus thus far of the introduction. Corrected 

56-58 Rephrase to make more concise. Unless introducing monetary value of the services, I’d rephrase to “importance”. Something like “The importance of ecosystem services supported by bivalve-built reefs is gaining recognition by government entities and industry.” It would be great if the authors could point to an example of this increased recognition, there are certainly numerous examples. Perhaps the authors could cite a few papers that lay out examples that support the sentence. When editing this sentence, I recommend the authors remove the last portion of the sentence “for the benefits…” since that is implied when mentioning ecosystem services in the beginning of the sentence. “In addition to providing physical niches for other species, biogenic shellfish reefs perform other provisioning, supporting and regulating roles [2][17][18][19]. These include bio-deposition [20][21] and providing essential fishery habitat [14][15]. The importance of ecosystem services supported by bivalve reefs are gaining recognition by government entities and industry [2][22]. This is also reflected by the increased investment by companies and NGOs into restoration projects. For example the Glenmorangie Whisky Company, in partnership with the Marine Conservation Society and Heriot-Watt University have established the Dornoch Environmental Enhancement Project [13][23]. This initiative aims to restore native oysters to the Dornoch Firth whilst enhancing biodiversity and water quality in the system: the oyster filter feeding is expected to account for organic waste from the distillery’s discharge water.” 61

59-61 Would benefit from references about the risks to shellfish reefs. Also surprised that climate change (and synergistic effects) not listed among the risks in the long-term Corrected 72

79 Rephrase to something like “Coincidence with the increased understanding of how important and vulnerable biogenic shellfish reefs are, actions to protect this now rare habitat are now also on the rise.” “Due to the increased understanding of how importance and vulnerable biogenic shellfish reefs are, actions to protect these now rare habitats are on the rise.” 90

81 I would clarify that oyster restoration is a relatively new approach--in terrestrial and (less so) marine habitats, it is not a new approach. If the authors choose to keep this sentence as is, I suggest clarifying the conservation/management strategy they are comparing restoration to that validates restoration as “relatively new”. I encourage the authors to re-write this sentence since I do not think it is accurate, as written. Worked in to beginning of the introduction. 35

83 Sometimes restoration plans include plans for chronic intervention (if necessary) so I would add “often” to the sentence--“This often requires building a self-sufficient…”. Additionally, a sentence on passive restoration to encourage natural recovery could be included here. Corrected 

85 Topic sentence is very similar to previous sentences above. I would rephrase. “The significance of restoration projects are now acknowledged at an international level with support from industry, non-governmental organisations and by governments” 97

93 Remove first clause of sentence, start paragraph with “Oyster restoration is…” Corrected 

98 Replace “articulate” with “re-construct” and revise sentence--“…it is extremely difficult to re-construct the biodiversity characteristics of a fully restored reef, given how rare pristine habitats have become” Corrected 

99-101 Rephrase to better transition from previous sentence. It’s my understanding that the authors are trying to state that studies that compare biodiversity in restored and less disturbed/somewhat natural habitats is necessary for measuring progress towards reaching biodiversity goals. I would re-write this last sentence to reflect this study goal, that will also lead into study aims nicely. “By examining the biodiversity of recovering oyster habitats this ecological understanding can be gained [2][41] and used to manage expectations of biodiversity outcomes and to measure progress towards biodiversity goals.” 109

106 Rewrite slightly--Something like…“The implications of this work extend from identifying likely expectations for biodiversity development post-restoration to the management of fished biogenic habitats and the biodiversity they support.” “The implications of this work extend from identifying likely expectations for biodiversity development during and after restoration activities to the management of fished biogenic habitats and the biodiversity they support.” 116

111 Could rephrase Hypothesis 2 to clarify language -- “Recovery of macrofaunal community biodiversity is influenced by time since disturbance”

Hypothesis Question: The authors have not defined biodiversity yet and this term is often overused or misused. It may be nice to define this in the introduction so the reader is aware ahead of time whether you are talking about biodiversity as species richness and evenness (using H’) or some other variable like species density, which is being used as a proxy for biodiversity.

Since biodiversity is a key part of this study, the authors could add more to the introduction and discussion as to why biodiversity is ecologically important. Provide specific examples, cite relevant literature, etc. The majority of the intro was focused on ecosystem services and the importance of biogenic habitats but the ecological importance of the diverse assemblage of species that these habitats support could be strengthened and made more prominent in the intro. “Biodiversity recovery of the macrofaunal community associated with an oyster reef is influenced by the length of time since disturbance.”

 121

115-117 The manuscript would benefit from a clearer description of the methods used in a few areas. Specifically:

Lines 115-117 (study site) - Suggest adding more site description. For instance - What are the depths of the reefs being surveyed? Other than a rotational system, how does the Loch Ryan Oyster Fishery Company operate (e.g., gear type, season, etc.)? “The fishery has been run by the Wallace family since 1701 and the fishery vessel uses a single dredge which only harvests the animals in the winter months, to avoid the spawning season. The oyster bed is divided into six plots which are harvested in a six-year rotation [42] and were regarded as treatments in the present study.” 127

118 I’d remove the term “plots” and only refer to them as Treatment 1, 2, and 3, each with four replicate transects. Ideally, the authors would have tracked each Treatment as it evolved since there may be differences between treatments unrelated to treatment (i.e., distance to shore, hydrology, relief of bedroom, depth, turbidity, etc.).

Authors should address the above (and other) potential confounding factors. If these factors were not evaluated during the study period, their omission should be acknowledged as a potential explanation that could weaken the strength of the findings. Looking at Figure 1 the Treatments are not interspersed and are clumped, so the potential confoundment of the results due purely to environmental factors and not treatment needs to be addressed (if the data was not collected and shown to be similar across treatments). If this environmental data exists, it should be highlighted in the section that outlines the details for all the various treatments.

Is there no reference habitat? Is Treatment 3 serving as the reference habitat? Since this was a prominent topic in the introduction, I suggest aligning the experimental design to the motivation outlined in the intro. “Sampling was conducted along a geographic gradient parallel to the shore where environmental conditions were not expected to significantly vary.”

“In keeping with most biodiversity studies, the sampling method presented here is expected to represent the patterns in the entirety of the community. Potential confounding environmental factors, were not included in the study. However as fishing is the dominant structural influence on the community the results from the study are out of proportion with any environmental gradients that would produce this scale of change. Furthermore, the scale and trend in the observed relationships are consistent with successional differences not spatial patterns.” 134

217

120-121 The four replicate transects in each plot are arranged quite differently (treatment 1 – in a rectangle, treatment 2 – in a line spread out, treatment 3 – clustered together). How were the locations of the four transects in each plot selected, and what if any impact might that have on the results? Do the characteristics of each transect vary at all? Co-ordinate mistake was made -> see updated figure 1 

124 Why were quadrats placed randomly instead of at some standardized interval? Could the authors elaborate on this? Were they opportunistically sampling along the transect line on whatever side had the most biogenic habitat? I suggest the authors justify the sampling design. “to avoid spatial auto-correlation” 139

123-125 Clearer description of the photo quadrat layout. The picture of the quadrats in figure 2 depicts a grid of 25 quadrats. Were a random sample of those selected at each location along the transect, or just one .5x.5m quadrat photographed/counted at each location, randomly selected along the 25m transect?

“These data were then averaged for each transect in each treatment.” 143

Figure 3 is missing axis labels/titles, making it hard to interpret MDS plot is a 2D spatial representation therefore axis are not normally included. If it is felt necessary the axis would only be labelled as “axis 1” & “axis 2”. 

142-143 I’ll defer to the other reviewers here to verify, but I don’t believe Pielou’s evenness should be run on transformed data. If the data were transformed for evaluating evenness, what was the justification? “…species abundance data were first square-root transformed before being used to calculate Shannon-Wiener’s diversity (H’) and Margalef’s richness (d) biodiversity indices.” 160

189-190 What is the relationship between shell cover and diversity/richness? Reported as strongly related but the correlation/significance is not specified. “were found to be positively correlated to both shell density and shell cover”

 206

Line 198 Where did the recovery trajectory of ten years come from? Is this from the literature (if so, needs citation) or the study? “with a recovery trajectory based on logarithmic models (Fig 5.) of approximately ten years” 215

198 How are the authors defining recovery trajectory? The curve plateauing? What if the plateau in the fished areas is still below what would have been the biodiversity “target” of a natural undisturbed oyster reef? This has not been addressed yet but is critical. If ten years is the “recovered” state, Fig 4 should extend to 10 like Fig 5 does since shell density, which covaries with time since harvesting, is stated as the ultimate cause of increased biodiversity. “Reef recovery likely begins as early as one year after fishing disturbance and increases over time as oyster spat settles and the consequential density and complexity of oyster shell increases. The recovery time (with sufficient larval supply) for O. edulis biogenic reefs (Figure 7; Figure 8) is potentially decadal in nature [26][47] because O. edulis as the biogenic ecosystem engineer, takes around three to five years to reach maturity [3]. Beyond the six year timeline examined here the biodiversity trajectory is expected to plateau out, but there is would be a need to examine oysters in an undisturbed sate to confirm this. Elsewhere in longer term marine reserves studies, ‘trophic cascades’ have occurred when re-established higher trophic level species have restructured benthic communities beyond simple biodiversity and biomass increases that were initially observed [53]. Although a six year timeline is too short to exclude either this or an ‘intermediate disturbance’ biodiversity pattern [54], as the reef is undergoing physical structural recovery these are unlikely.” 239

200 The authors have yet to acknowledge that what they may be witnessing is a version of the intermediate disturbance hypothesis. What if natural reefs have lower biodiversity? It’s important to acknowledge that the 10-year “recovery” is based on a disturbed system. The authors should include text that acknowledges 1) no reference included, 2) confounding factors could explain differences in biodiversity, 3) have not tracked individual treatments over time, only a snapshot in time so can’t rule out #2. See above response. 239

206 I don’t think this is an issue at all actually since the level of detail in sampling was the same across all treatments. “In keeping with most biodiversity studies, the sampling method presented here is expected to represent the patterns in the entirety of the community.” 219

213-215 Minor clarification, would suggest adding "after 6 years" after "allowing macrofauna biodiversity to recover by 60%". As currently written, could be misinterpreted to mean that biodiversity recovers by 60% after each fishing season. Corrected 

224 Why inevitably? “is potentially decadal in nature” 242

225 What is several years? 2-3 or 5-6? Please specify. “takes around three to five years” 243

223-225 Is this where the ten years comes from? 

223-225 The study results show that recovery begins as early as 1 year post-harvest and increases over time as shell density increases, but without knowing what biodiversity would be without any harvest, it’s unclear that the recovery would be on a ten year cycle when the fished system is on a 6-year cycle. Oysters would be recruiting throughout that 6-year period and the contributions of each cohort to overall shell density would be staggered. It’s also possible there is a benefit to biodiversity when the size structure of the oyster reef is varied and that’s what’s driving the pattern (another confounding factor that should be mentioned in the discussion). Treatment 3 is going to have a broader size frequency distribution than Treatment 2 or 1. Since this data was not collected, the contributions of a reef composed of different sized individuals should be addressed.

Figure comment: Have symbols match across all figures, when applicable. “The recovery of bivalves is dependent on successful larval settlement, recruitment, and growth, suggesting the need to also maintain overall adult stock levels as a source in any restoration or sustainable management scenario. Reef recovery likely begins as early as one year after fishing disturbance and increases over time as oyster spat settles and the consequential density and complexity of oyster shell increases. The recovery time (with sufficient larval supply) for O. edulis biogenic reefs (Figure 7; Figure 8) is potentially decadal in nature [26][47] because O. edulis as the biogenic ecosystem engineer, takes around three to five years to reach maturity [3].” 237

223 remove “aspects of the” Corrected 

237-240 Could this be due to the down-weighting of abundances when transformed? Please clarify whether the data used for the evenness index was transformed. If it was transformed, justify. If it was not transformed, include text that distinguished what data was vs was not transformed. Corrected 

261 Is 1701 the age of the Loch Ryan Company? If so, that detail should be in Intro or Methods. Corrected 

Data availability Data on percent oyster shell cover were not available, but the remaining data were. Updated supporting data 

In addition to the above comments, all spelling and grammatical errors pointed out by the

reviewers have been corrected.

We look forward to hearing from you in due time regarding our submission and to respond to

any further questions and comments you may have.

Sincerely,

Naomi Kennon

05/09/22

---

## [Decision Letter · Decision Letter 1]

4 Oct 2022

PONE-D-22-14818R1Rotational fishing enables biodiversity recovery and provides a model for oyster (*Ostrea edulis*) habitat restorationPLOS ONE

Dear Dr. Kennon,

Thank you for submitting your manuscript to PLOS ONE. After careful consideration, we feel that it has merit but does not fully meet PLOS ONE’s publication criteria as it currently stands. Therefore, we invite you to submit a revised version of the manuscript that addresses the points raised during the review process.

We look forward to receiving your revised manuscript.

Kind regards,

José A. Fernández Robledo, Ph.D.

Academic Editor

PLOS ONE

Journal Requirements:

Reviewers' comments:

Reviewer's Responses to Questions

**Comments to the Author**

1. If the authors have adequately addressed your comments raised in a previous round of review and you feel that this manuscript is now acceptable for publication, you may indicate that here to bypass the “Comments to the Author” section, enter your conflict of interest statement in the “Confidential to Editor” section, and submit your "Accept" recommendation.

Reviewer #2: All comments have been addressed

2. Is the manuscript technically sound, and do the data support the conclusions?

Reviewer #2: Yes

3. Has the statistical analysis been performed appropriately and rigorously? 

Reviewer #2: Yes

4. Have the authors made all data underlying the findings in their manuscript fully available?

Reviewer #2: Yes

5. Is the manuscript presented in an intelligible fashion and written in standard English?

Reviewer #2: Yes

6. Review Comments to the Author

Reviewer #2: The authors addressed all of the comments and suggested edits. I have a few remaining comments and suggestions that should be addressed. Most of these revisions focus on improving language and clarity of the main text.

FIGURE 2 - consent for photos is typically for those pictured in the photo. I believe PLOS ONE policy is that those scientific divers submit a consent form for their photo to be included in the paper. An exception to this may be if the scientific divers are also co-authors on the manuscript. I would defer to the Editor on this though.

Line 13: Rephrase, shouldn’t end the sentence with “in”. I suggest, “European oyster habitat are now considered rare and there are few examples of undisturbed reefs from which restoration sites could be compared.”

Line 15: I suggest rephrasing--I don’t consider oyster restoration a “feature” or marine conservation practice. Perhaps rephrase to…”…oyster restoration efforts are on the rise, becoming a more prominent component of Europe’s portfolio for marine conservation practices” or something along those lines?

Revised Line 35: I appreciate the authors editing the introduction to make it more applicable to a broader audience. I have indicated line edits in all caps to the revised text below. Deleted “making them rare or endangered”. The authors could emphasize that rarely do we have the data or historical knowledge of the pre-degraded state to set a desired target for a restored habitat. Often, the best we can do is compare restored habitats to ambient conditions of nearby, less impacted, sites (reference sites). I suggest the author re-phrase the third sentence--it’s often a lack of data (not only studies) that make it difficult to determine what condition/historical state you are targeting for restoration. I attempted to provide an option for revision to better connect the last two sentences in the revised paragraph, the authors should review and consider how to best communicate their point here which I believe to be that 1) success requires a fundamental understanding of how a particular ecosystem functions, 2) we lack the data necessary to understand how a restored habitat should function given many of these habitats are rare or degraded and don’t function how they “should”, 3) the best we can do is compare a restored habitat to a nearby reference habitat that for the location, local stressors, etc. serves as the most appropriate target for evaluating restoration outcomes.

“Habitat and biodiversity restoration is becoming an increasingly popular conservation practice around the world in both terrestrial and aquatic habitats [1][2][4][5]. Many of these habitats have been lost or degraded due to anthropogenic activities [3]. EFFECTIVE RESTORATION REQUIRES an ecological understanding of the TARGETED habitat, but in many cases, due to the degradation or rarity of the habitat, WE LACK THE DATA NEEDED TO DETERMINE THE DESIRED CONDITION AND ECOLOGICAL FUNCTIONING FOR A RESTORED HABITAT. A reference habitat (analogous in terms of community composition, niche fulfilment and provision of ecosystem services)[6] can provide a baseline against which restorative progress can be judged and expectations of biodiversity outcomes clarified (see [7]).”

Line 48: Common since when? I would rephrase this sentence. It could be made more clear. I would start with something like “In Europe, native oyster restoration projects have been increasing over the past XYZ”.

By providing the approximate time from which oyster restorations became more common, the authors will get at their earlier point in their original draft that it is a relatively new field that lacks data.

Line 50: Composition of what? Species composition? How is this different from biodiversity in many ways they are components of one another. Perhaps add in a reference that highlights the lack of knowledge on other reef functions (i.e. water filtration, habitat provisioning) and replace “composition” with whatever function you’d like to highlight along with biodiversity.

Line 69-70: Replace “to account for” with “mitigate the impacts of organic waste from the distillery’s discharge water”.

Line 90: Revise. “importance” should be “important”

Line 109: Revise. The revised sentence is still disjointed and hard to follow. What is an ‘ecological understanding’? The authors use this phrase frequently throughout but are mostly concerned with biodiversity. The authors could clarify by better communicating the link between biodiversity and ecosystem functioning. The language is very broad here and should be revised.

Line 139: Authors should state in the methods that the sampling was opportunistic.

Line 143: A supplemental figure of the transect survey layout would be helpful.

Line 217: “Furthermore, the scale and trend in the observed relationships are consistent with successional differences (ADD CITATION HERE) not spatial patterns.”

Figure 3: Add axis 1 and axis 2 labels as suggested by authors.

Line 217: List/acknowledge potential confounding factors that were not co-monitored in this sentence “Potential confounding environmental factors (LIST HERE), were not included in the study.”.

Line 217: Explain what the authors mean by “out of proportion” with any environmental gradients? Also, this added language makes many sweeping statements without acknowledging what environmental gradients, factors, etc. may be at play. The authors should add specificity to the main text about the potential confounding factors that they did not evaluate.

Line 219: Add citations after “In keeping with most biodiversity studies,…”

Line 239: Remove the word “consequential”. Revise “is potentially decadal in nature”--this language could be much improved. The authors should revise this section slightly for conciseness. The data supports that reef recovery begins within a year post disturbance and reaches reference levels within ten years.

7. PLOS authors have the option to publish the peer review history of their article (what does this mean?). If published, this will include your full peer review and any attached files.

Reviewer #2: **Yes: **Kathryn Beheshti

---

## [Author Response · Author response to Decision Letter 1]

3 Nov 2022

Thank you for giving me a second opportunity to submit a revised draft of my manuscript titled Rotational fishing enables biodiversity recovery and provides a model for oyster (Ostrea edulis) habitat restoration to Plos One. 

We again appreciate the time and effort that you and the reviewers have dedicated to providing your valuable feedback on my manuscript. We are grateful to the reviewers for their insightful comments on my paper and have been able to incorporate changes to reflect the suggestions provided by the reviewers. 

See the following table for a point-by-point response to the reviewers’ comments and concerns.

Figure 2 Consent for photos is typically for those pictured in the photo. I believe PLOS ONE policy is that those scientific divers submit a consent form for their photo to be included in the paper. An exception to this may be if the scientific divers are also co-authors on the manuscript. I would defer to the Editor on this though. Divers are authors, included initials in caption. n/a

13 Rephrase, shouldn’t end the sentence with “in”. I suggest, “European oyster habitat are now considered rare and there are few examples of undisturbed reefs from which restoration sites could be compared.” Revision accepted 13

35 I appreciate the authors editing the introduction to make it more applicable to a broader audience. I have indicated line edits in all caps to the revised text below. Deleted “making them rare or endangered”. The authors could emphasize that rarely do we have the data or historical knowledge of the pre-degraded state to set a desired target for a restored habitat. Often, the best we can do is compare restored habitats to ambient conditions of nearby, less impacted, sites (reference sites). I suggest the author re-phrase the third sentence--it’s often a lack of data (not only studies) that make it difficult to determine what condition/historical state you are targeting for restoration. I attempted to provide an option for revision to better connect the last two sentences in the revised paragraph, the authors should review and consider how to best communicate their point here which I believe to be that 1) success requires a fundamental understanding of how a particular ecosystem functions, 2) we lack the data necessary to understand how a restored habitat should function given many of these habitats are rare or degraded and don’t function how they “should”, 3) the best we can do is compare a restored habitat to a nearby reference habitat that for the location, local stressors, etc. serves as the most appropriate target for evaluating restoration outcomes. “Habitat and biodiversity restoration is becoming an increasingly popular conservation practice around the world in both terrestrial and aquatic habitats [1][2][4][5]. Many of these habitats have been lost or degraded due to anthropogenic activities [3]. EFFECTIVE RESTORATION REQUIRES an ecological understanding of the TARGETED habitat, but in many cases, due to the degradation or rarity of the habitat, WE LACK THE DATA NEEDED TO DETERMINE THE DESIRED CONDITION AND ECOLOGICAL FUNCTIONING FOR A RESTORED HABITAT. A reference habitat (analogous in terms of community composition, niche fulfilment and provision of ecosystem services)[6] can provide a baseline against which restorative progress can be judged and expectations of biodiversity outcomes clarified (see [7]).”

 Rephrased: “Habitat and biodiversity restoration is becoming an increasingly popular conservation practice around the world in both terrestrial and aquatic habitats [1][2][3][4]. Many of these habitats have been lost or degraded due to anthropogenic activities [5], making them rare or endangered. Effective restoration requires an ecological understanding of the targeted habitat, but in many cases, we lack the data needed to determine the desired condition and biodiversity characteristics for a restored habitat. A reference habitat (analogous in terms of community composition, niche fulfilment and provision of ecosystem services)[6] can provide a baseline against which restorative progress can be judged and expectations of biodiversity outcomes clarified (see [7]).”

(Following revision for line 109, the term ‘ecological functioning’ has been replaced with ‘biodiversity characteristics’ to improve clarity.) 

 44

48 Common since when? I would rephrase this sentence. It could be made more clear. I would start with something like “In Europe, native oyster restoration projects have been increasing over the past XYZ”. By providing the approximate time from which oyster restorations became more common, the authors will get at their earlier point in their original draft that it is a relatively new field that lacks data. 

 Rephrased: “In the last decade, restoration of the European native oyster (Ostrea edulis) has become a common feature in coastal conservation projects in Europe[1][8].” 51

50 Composition of what? Species composition? How is this different from biodiversity in many ways they are components of one another. Perhaps add in a reference that highlights the lack of knowledge on other reef functions (i.e. water filtration, habitat provisioning) and replace “composition” with whatever function you’d like to highlight along with biodiversity. 

 Rephrased: “Due to the rarity of intact O. edulis reef habitats (e.g., [8][9][10][11]) there is a lack of data with regards to the biodiversity and structure of the faunal community associated with the reefs [2]. It has therefore been necessary to look for reference habitats, which are analogous in terms of community composition, niche fulfilment and provision of ecosystem services. This allows for a baseline comparison that restorative progress can be judged and expectations of biodiversity outcomes clarified (see [7]).” 53

69-70 Replace “to account for” with “mitigate the impacts of organic waste from the distillery’s discharge water”. Revision accepted. 77

109 Revise. The revised sentence is still disjointed and hard to follow. What is an ‘ecological understanding’? The authors use this phrase frequently throughout but are mostly concerned with biodiversity. The authors could clarify by better communicating the link between biodiversity and ecosystem functioning. The language is very broad here and should be revised. Rephrased: “Oyster restoration is becoming a priority in many coastal conservation initiatives [14][8]. Many locations around Europe which supported historic oyster fisheries and /or remnant populations are now being considered as possible locations for O. edulis restoration [1][38]. Owing to the catastrophic decline and modern-day rarity of O. edulis reef habitats, and the historical context within which these declines occurred, it is extremely difficult to re-construct the biodiversity characteristics of a fully restored reef [39][40]. By examining the biodiversity of oyster habitats recovering from fishing activity, a greater understanding of the faunal community composition associated with oyster reef habitats can be gained [2][41] and used to manage expectations of biodiversity outcomes and progress towards biodiversity goals.” 

 113

139 Authors should state in the methods that the sampling was opportunistic. 

 Rephrased: “Three treatments were selected opportunistically for survey in 2019…” 142

143 A supplemental figure of the transect survey layout would be helpful. Revision accepted see supplemental figure 1. 149

217 “Furthermore, the scale and trend in the observed relationships are consistent with successional differences (ADD CITATION HERE) not spatial patterns.”

 Revision accepted. 234

Fig 3 Add axis 1 and axis 2 labels as suggested by authors. Revision accepted n/a

217 List/acknowledge potential confounding factors that were not co-monitored in this sentence “Potential confounding environmental factors (LIST HERE), were not included in the study.”. Rephrased: “Potential confounding environmental factors, were not included in the study (eg depth, temperature and tidal flow rate) because they were judged to be diminutive. Fishing is the dominant structural influence on the community, hence the results from the study are out of proportion with any environmental gradients that would produce this scale of change...” 230

219 Add citations after “In keeping with most biodiversity studies,…” Revision accepted. 228

239 Remove the word “consequential”. Revise “is potentially decadal in nature”--this language could be much improved. The authors should revise this section slightly for conciseness. The data supports that reef recovery begins within a year post disturbance and reaches reference levels within ten years. Revision accepted.

Rephrased: “the recovery time for O. edulis reefs (Figure 7; Figure 8) inevitably takes several years because the structure of the reef is dependent on established adult sized shellfish [26][47] [3]. Beyond the six-year timeline examined here the biodiversity trajectory is expected to plateau out, but examination of oysters in an undisturbed state would be required to confirm this.”

 254

In addition to the above comments, to improve readability and accuracy of statement two further changes have been made 

Line 93: Insert of term “shellfish reefs”

Line 94: deletion of term “late”

We look forward to hearing from you in due time regarding our submission and to respond to

any further questions and comments you may have.

Sincerely,

Naomi Kennon

---

## [Editor Report · Decision Letter 2]

9 Nov 2022

PONE-D-22-14818R2Rotational fishing enables biodiversity recovery and provides a model for oyster (Ostrea edulis) habitat restorationPLOS ONE

Dear Dr. Kennon,

Thank you for submitting your manuscript to PLOS ONE. After careful consideration, we feel that it has merit but does not fully meet PLOS ONE’s publication criteria as it currently stands. Therefore, we invite you to submit a revised version of the manuscript that addresses the points raised during the review process.

We look forward to receiving your revised manuscript.

Kind regards,

José A. Fernández Robledo, Ph.D.

Academic Editor

PLOS ONE

Journal Requirements:

Additional Editor Comments:

Dear Dr. Kennon,

please proceed addressing the reviewer's concerns and comments.

Sincerely,

José A. Fernández Robledo

AE

PLoS ONE
---

## [Author Response · Author response to Decision Letter 2]

14 Dec 2022

Dear José A. Fernández Robledo, Ph.D. 10/12/22 

Thank you for giving us an opportunity to submit a revised draft of our manuscript Rotational fishing enables biodiversity recovery and provides a model for oyster (Ostrea edulis) habitat restoration to PLOS One. 

We again appreciate the time and effort that you and the reviewers have dedicated to providing your valuable feedback on my manuscript. We are grateful to the reviewers for their insightful comments on the paper and have been able to incorporate changes to reflect the suggestions provided by the reviewers. 

See the following table for a point-by-point response to the reviewers’ comments and concerns.

Line Comment Response New line

Figure 2 Consent for photos is typically for those pictured in the photo. I believe PLOS ONE policy is that those scientific divers submit a consent form for their photo to be included in the paper. An exception to this may be if the scientific divers are also co-authors on the manuscript. I would defer to the Editor on this though. Divers are the authors. Relevant authors initials have been included in the figure caption. Fig 2 

13 Rephrase, shouldn’t end the sentence with “in”. I suggest, “European oyster habitat are now considered rare and there are few examples of undisturbed reefs from which restoration sites could be compared.” The reviewers point is accepted. However, insertion of “few examples of” does not represent the situation know to the authors, who don’t believe any examples are known. We have dealt with the reviewers point by removing “in”. Overall, we feel that this frames the present study appropriately. 13

15 I suggest rephrasing--I don’t consider oyster restoration a “feature” or marine conservation practice. Perhaps rephrase to…”…oyster restoration efforts are on the rise, becoming a more prominent component of Europe’s portfolio for marine conservation practices” or something along those lines? Revision accepted, rephrased to: “As more is understood of the ecosystem services provided by the reefs, oyster restoration efforts are on the rise, becoming a more prominent component of Europe’s portfolio for marine conservation practices.” 15

35 I appreciate the authors editing the introduction to make it more applicable to a broader audience. I have indicated line edits in all caps to the revised text below. Deleted “making them rare or endangered”. The authors could emphasize that rarely do we have the data or historical knowledge of the pre-degraded state to set a desired target for a restored habitat. Often, the best we can do is compare restored habitats to ambient conditions of nearby, less impacted, sites (reference sites). I suggest the author re-phrase the third sentence--it’s often a lack of data (not only studies) that make it difficult to determine what condition/historical state you are targeting for restoration. I attempted to provide an option for revision to better connect the last two sentences in the revised paragraph, the authors should review and consider how to best communicate their point here which I believe to be that 1) success requires a fundamental understanding of how a particular ecosystem functions, 2) we lack the data necessary to understand how a restored habitat should function given many of these habitats are rare or degraded and don’t function how they “should”, 3) the best we can do is compare a restored habitat to a nearby reference habitat that for the location, local stressors, etc. serves as the most appropriate target for evaluating restoration outcomes. “Habitat and biodiversity restoration is becoming an increasingly popular conservation practice around the world in both terrestrial and aquatic habitats [1][2][4][5]. Many of these habitats have been lost or degraded due to anthropogenic activities [3]. EFFECTIVE RESTORATION REQUIRES an ecological understanding of the TARGETED habitat, but in many cases, due to the degradation or rarity of the habitat, WE LACK THE DATA NEEDED TO DETERMINE THE DESIRED CONDITION AND ECOLOGICAL FUNCTIONING FOR A RESTORED HABITAT. A reference habitat (analogous in terms of community composition, niche fulfilment and provision of ecosystem services)[6] can provide a baseline against which restorative progress can be judged and expectations of biodiversity outcomes clarified (see [7]).”

 Revision accepted, rephrased to: “Habitat and biodiversity restoration is becoming an increasingly popular conservation practice around the world in both terrestrial and aquatic habitats [1][2][3][4]. Many of these habitats have been lost or degraded due to anthropogenic activities [5], making them rare or endangered. Effective restoration requires an ecological understanding of the targeted habitat, but in many cases, we lack the data needed to determine the desired condition and biodiversity characteristics for a restored habitat. A reference habitat (analogous in terms of community composition, niche fulfilment and provision of ecosystem services)[6] can provide a baseline against which restorative progress can be judged and expectations of biodiversity outcomes clarified (see [7]).”

(Following revision for line 109, the term ‘ecological functioning’ has been replaced with ‘biodiversity characteristics’ to improve clarity.) 

 44

48 Common since when? I would rephrase this sentence. It could be made more clear. I would start with something like “In Europe, native oyster restoration projects have been increasing over the past XYZ”. By providing the approximate time from which oyster restorations became more common, the authors will get at their earlier point in their original draft that it is a relatively new field that lacks data. 

 Revision accepted, rephrased to: “In the last decade, restoration of the European native oyster (Ostrea edulis) has become a common component of coastal conservation projects in Europe[1][8].” 51

50 Composition of what? Species composition? How is this different from biodiversity in many ways they are components of one another. Perhaps add in a reference that highlights the lack of knowledge on other reef functions (i.e. water filtration, habitat provisioning) and replace “composition” with whatever function you’d like to highlight along with biodiversity. 

 Revision accepted, rephrased to: “Due to the rarity of intact O. edulis reef habitats (e.g., [8][9][10][11]) there is a lack of data with regards to the biodiversity and structure of the faunal community associated with the reefs [2].” 53

69-70 Replace “to account for” with “mitigate the impacts of organic waste from the distillery’s discharge water”. Revision accepted, rephrased to: “This initiative aims to restore native oysters to the Dornoch Firth whilst enhancing biodiversity and water quality in the system: the oyster filter feeding is expected to mitigate organic waste from the distillery’s discharge water.” 77

90 Revise. “importance” should be “important” Revision accepted. 100

109 Revise. The revised sentence is still disjointed and hard to follow. What is an ‘ecological understanding’? The authors use this phrase frequently throughout but are mostly concerned with biodiversity. The authors could clarify by better communicating the link between biodiversity and ecosystem functioning. The language is very broad here and should be revised. Revision accepted, rephrased to: “Oyster restoration is becoming a priority in many coastal conservation initiatives [14][8][35][36]. Many locations around Europe which supported historical oyster fisheries and /or remnant populations are now being considered as possible locations for O. edulis restoration [1][37][38]. Owing to the catastrophic decline and modern-day rarity of O. edulis reef habitats, and the historical context within which these declines occurred, it is extremely difficult to re-construct the biodiversity characteristics of a fully restored reef [39]. By examining the biodiversity of oyster habitats recovering from fishing activity[40], a greater understanding of the faunal community composition associated with oyster reef habitats can be gained [2][10][31] and used to manage expectations of biodiversity outcomes and progress towards biodiversity goals.” 

 113

139 Authors should state in the methods that the sampling was opportunistic. 

 Revision accepted, rephrased to: “Three treatments were selected opportunistically for survey in 2019…” 142

143 A supplemental figure of the transect survey layout would be helpful. Revision accepted see S1 Figure. In-text citation added to line 148. 148

217 “Furthermore, the scale and trend in the observed relationships are consistent with successional differences (ADD CITATION HERE) not spatial patterns.”

 Revision accepted. Added the following references: [10] Smyth & Roberts 2010 [25]Kaiser et al., 2006 [46] Scriberras et al., 2018 [47] Farinas-Franco & Roberts 2014 234

Fig 3 Add axis 1 and axis 2 labels as suggested by authors. Revision accepted Fig 3

217 List/acknowledge potential confounding factors that were not co-monitored in this sentence “Potential confounding environmental factors (LIST HERE), were not included in the study.”. Revision accepted, rephrased to: “Potential confounding environmental factors were not included in the study (eg depth, temperature and tidal flow rate) because they were judged to be diminutive.” 230

217 Explain what the authors mean by “out of proportion” with any environmental gradients? Also, this added language makes many sweeping statements without acknowledging what environmental gradients, factors, etc. may be at play. The authors should add specificity to the main text about the potential confounding factors that they did not evaluate. Revision accepted, rephrased to: “Fishing is the dominant structural influence on the community, hence the results from the study are unlikely to be the result of environmental gradients that could produce such a significant scale of change. Furthermore, the scale and trend in the observed relationships are consistent with successional differences [10][25][46][47] not spatial patterns” 234

219 Add citations after “In keeping with most biodiversity studies,…” Revision accepted. Added the following references: [44]Kent et al., 2017 [45] Benjamin et al., 2022 228

239 Remove the word “consequential”. Revise “is potentially decadal in nature”--this language could be much improved. The authors should revise this section slightly for conciseness. The data supports that reef recovery begins within a year post disturbance and reaches reference levels within ten years. Revision accepted, rephrased to: “Reef recovery likely begins as early as one year after fishing disturbance and increases over several more years as oyster spat settles and the size, density and complexity of oyster shell increases (Figure 8; Figure 9) [3][26][47]. Beyond the six-year timeline studied here the biodiversity trajectory is expected to plateau out, but examination of oysters in an undisturbed state would be required to confirm this.” 

 254

In addition to the above comments, to improve readability and accuracy of statement further minor changes have been made: 

Line 74: replacement of the word “is” with “are”

Line 93: Insert of term “shellfish reefs”

Line 94: deletion of word “late”

Line 107: replacement of the word “is” with “are”

Line 296: replacement of the word “Therefore” with “Overall”

Line 298: deletion of word “Overall”

In the acknowledgments line 134 has been added to reflect our gratitude for the referees feedback: “The comments from the referees greatly improved the manuscript”.

Furthermore, we have revisited text in lines 256-263 concerning intermediate disturbance and trophic cascades. We feel that the new edits more clearly address the reviewers’ previous comments and cover the topic more accurately. The section of text has been rephrased to: “Elsewhere in longer term (10yr+) marine reserves studies, ‘trophic cascades’ have occurred when re-established higher trophic level species have restructured benthic communities beyond simple biodiversity and biomass increases that were initially observed [eg. 55]. A six year timeline in the present study is too short to exclude the possibility that trophic cascades might subsequently occur in oyster reef recovery. Attributing enhanced biodiversity to ‘intermediate disturbance’ [56] in the present study, however, seems unlikely because increasing niches as the reef undergoes physical structural recovery are unlikely to abate.”

The following references were added to support the suggested revisions from the reviewer: Farinas-Franco & Roberts 2014 to line 217; Kent et al., 2017 and Benjamin et al., 2022 to line 219.

Likewise, references have been omitted from the manuscript because, due to revisions, they were no longer relevant: Haddad et al., 2015 from line 99; Graham & Nash 2013 from line 106 and 274. 

Due to the changes to the references the in-text citations have been updated.

Finally a list of the supporting information captions has been added to the end of the manuscript in addition to relevant in-text citations and all figures have been processed through PACE and now meet PLOS requirements.

We look forward to hearing from you in due time regarding our submission and to respond to

any further questions and comments you may have.

Sincerely,

Naomi Kennon

10/12/22

---

## [Decision Letter · Decision Letter 3]

26 Dec 2022

PONE-D-22-14818R3Rotational fishing enables biodiversity recovery and provides a model for oyster (Ostrea edulis) habitat restorationPLOS ONE

Dear Dr. Kennon,

Thank you for submitting your manuscript to PLOS ONE. After careful consideration, we feel that it has merit but does not fully meet PLOS ONE’s publication criteria as it currently stands. Therefore, we invite you to submit a revised version of the manuscript that addresses the points raised during the review process.

We look forward to receiving your revised manuscript.

Kind regards,

José A. Fernández Robledo, Ph.D.

Academic Editor

PLOS ONE

Journal Requirements:

Additional Editor Comments (if provided):

Dear Dr. Kennon,

as the reviewer mentioned, they are a few minor edits that need to be addressed.

Happy Holidays.

-j

Reviewers' comments:

Reviewer's Responses to Questions

**Comments to the Author**

1. If the authors have adequately addressed your comments raised in a previous round of review and you feel that this manuscript is now acceptable for publication, you may indicate that here to bypass the “Comments to the Author” section, enter your conflict of interest statement in the “Confidential to Editor” section, and submit your "Accept" recommendation.

Reviewer #2: All comments have been addressed

2. Is the manuscript technically sound, and do the data support the conclusions?

Reviewer #2: Yes

3. Has the statistical analysis been performed appropriately and rigorously? 

Reviewer #2: Yes

4. Have the authors made all data underlying the findings in their manuscript fully available?

Reviewer #2: Yes

5. Is the manuscript presented in an intelligible fashion and written in standard English?

Reviewer #2: Yes

6. Review Comments to the Author

Reviewer #2: The revised manuscript is much improved--great work. I appreciate how the authors approached edits suggested by my previous reviews. This paper is going to be a valuable contribution to the field of oyster conservation and management. That said, I had a suite of minor revisions I suggest the authors consider. My final suggested edits and comments are below.

Abstract: In the last sentence--“indicate the likely scale of biodiversity gain” is unclear. The authors could consider replacing with “indicate the probable biodiversity benefits of oyster habitat restoration…”. The authors should also replace “tool” with “metric”, as shell density is a metric not a tool.

Line 40: Rephrase; “Habitat restoration to enhance biodiversity is becoming…”

Line 42: Delete “making them rare or endangered” or add a citation.

Line 72: Replace “bottom-contacting fishing gear” with “Bottom towed fishing gear” to match the following sentence.

Line 74: If correct, the better reflect the destruction of these practices, could consider replacing “some” with “many”.

Line 77: Replace “with” with “and a significant reduction in biodiversity”

Lines 79-81: Need citation

Line 85: Replace “around” with “in”

Line 92: Replace “disruption” with “disturbance”

Lines 97-98: Remove “From 2021” and start the sentence with “The UN Decade on…”

Line 101: Replace restoration projects with restoration practitioners since you’re naming entities/stakeholders in this sentence.

Line 119: Hypothesis 1 does not address the live vs. dead shell component described above. This should be integrated into the hypothesis.

Line 135: I recommend removing “geographic gradient” as this would suggest there are environmental conditions that vary along that gradient (different reef rigosity, depth, etc. could impact flow, temperature, light penetration). If the authors are convinced that environmental conditions were similar across transects, it’d be best to remove “geographic gradient” and have the sentence read as “Sampling was conducting along X (put number of transects here) 25 m transects that ran parallel to shore.”

Lines 143-145: Did the divers count the shells in the field or was all counting down using ImageJ--please clarify.

Line 154: Replace “within” with “by”

Line 164: Should MDS be nMDS?

Lines 172-173: Rephrase to - (“diminishing rate of increase as reef approaches ‘recovered’ condition”)

Line 173: The authors should explain why year was log-transformed.

Lines 177-178: Should be in methods

Line 181: Replace “a very good” with “is an accurate representation”

Line 192: Table 1 -- Referring now to the treatments and treatment sites is confusing to the reader, I suggest maintaining the naming system above and referring to Treatments 1 and 3 (removing “site”).

Lines 195-197: It’d be great if in the methods, the authors wrote a quick few sentences about the strengths and weaknesses of Shannon-Wieners, Margalef’s and Pielou’s evenness so the reader can better make sense of the results and what each of these values means.

Line 198: Average oyster shell density increased between what treatments? Only 1 and 3? Need to specify.

Line 217: Rephrase--“…is expected to represent community-wide patterns”

Line 218: This is great, I think it’s really important to acknowledge the data that wasn’t collected for this study and could have (but are unlikely to have) influenced results. Instead of “included in the study” replace with “were not evaluated”.

Line 230; 233: Replace bottom-contacting with bottom towed. Unless the authors are referring to other fishing practices (shrimp pots, lobster traps, bottom trawling)--if this is the case, some examples in parentheses would be helpful.

Line 246: Rephrase to “…that trophic cascades could ultimately transform the benthic community and affect oyster reef recovery”

Line 247-249: I suggest the authors improve the text in this sentence here as it’s unclear what they are trying to say is the reasoning behind why their study is not evidence of intermediate disturbance hypothesis. Also, intermediate disturbance hypothesis appears out of nowhere so perhaps the authors could add a sentence before the current one that addresses other potential explanations to support their findings and then follow it up with why it doesn’t apply (as they’ve done, but with slight editing to make clearer).

Line 250: What kind of relationship? Positive? Negative? This type of information is important for guiding the reader.

Lines 250-258: The authors should indicate the directionality of all of the relationships in this paragraph to help guide the reader.

Lines 260-263: Great!

Line 278: Rephrase slightly: “perhaps the most direct evidence yet quantifying the potential biodiversity gain…”

Line 280: Replace “would probably double in a decade” to “would likely double within a decade”

Line 281: Make more concise; “the present findings demonstrate the benefits of restoration and the trajectory of restoration success as it relates to biodiversity.” Then, the authors could start a new sentence “Our data showed a link between increased shell material and increased biodiversity.”

Line 283: Missing a period.

Line 284: Rephrase; “rare example from a long-term dredge fishery whose practices appear to have allowed rare oyster habitat and the associated community to persist, thereby providing a valuable insight into the recovery of biodiversity by balancing sustainable fishery practices with European oyster habitat recovery efforts”. Since this study did not track restoration over time but recovery post disturbance, distinguishing between the two is critical.

7. PLOS authors have the option to publish the peer review history of their article (what does this mean?). If published, this will include your full peer review and any attached files.

Reviewer #2: No

---

## [Author Response · Author response to Decision Letter 3]

26 Jan 2023

Dear José A. Fernández Robledo, Ph.D. 24/01/23 

Thank you for giving us an opportunity to submit a revised draft of our manuscript Rotational fishing enables biodiversity recovery and provides a model for oyster (Ostrea edulis) habitat restoration to PLOS One. 

We again appreciate the time and effort that you and the reviewers have dedicated to providing valuable feedback on our manuscript. We are grateful to the reviewers for their insightful comments on the paper and have been able to incorporate these further changes to reflect the suggestions provided. 

See the following table for a point-by-point response to the reviewers’ comments and concerns.

Line Comment Response New line

Abstract: In the last sentence “indicate the likely scale of biodiversity gain” is unclear. The authors could consider replacing with “indicate the probable biodiversity benefits of oyster habitat restoration…”. The authors should also replace “tool” with “metric”, as shell density is a metric not a tool. Revision accepted. Rephrased to “The findings from the present study indicate the probable biodiversity benefits of oyster habitat restoration and a cost-effective metric (shell density) to judge progress in restoration projects” 36

40 Rephrase; “Habitat restoration to enhance biodiversity is becoming…” Revision accepted. Rephrased to “Habitat restoration to enhance biodiversity is becoming an increasingly popular conservation practice” 41

42 Delete “making them rare or endangered” or add a citation. Revision accepted; line deleted. 43

72 Replace “bottom-contacting fishing gear” with “Bottom towed fishing gear” to match the following sentence. Revision accepted. Rephrased to “towed” 74

74 If correct, the better reflect the destruction of these practices, could consider replacing “some” with “many”. Revision accepted. Rephrased to “damages and kills many benthic organisms” 76

77 Replace “with” with “and a significant reduction in biodiversity” Revision accepted. Rephrased to “and a significant reduction in biodiversity” 80

79-81 Need citation Added [2][16][17][26]. 83

85 Replace “around” with “in” Revision accepted. 87

92 Replace “disruption” with “disturbance” Revision accepted. 94

97-98 Remove “From 2021” and start the sentence with “The UN Decade on…” Revision accepted. 100

101 Replace restoration projects with restoration practitioners since you’re naming entities/stakeholders in this sentence. Revision accepted. 104

119 Hypothesis 1 does not address the live vs. dead shell component described above. This should be integrated into the hypothesis. Revision accepted. Rephrased to “The macrofaunal biodiversity of an oyster reef community is affected by the density of live and dead oyster shells.” 124

135 I recommend removing “geographic gradient” as this would suggest there are environmental conditions that vary along that gradient (different reef rigosity, depth, etc. could impact flow, temperature, light penetration). If the authors are convinced that environmental conditions were similar across transects, it’d be best to remove “geographic gradient” and have the sentence read as “Sampling was conducting along X (put number of transects here) 25 m transects that ran parallel to shore.” Revision accepted. Rephrased to “Sampling was conducted along a total of twelve 25m transects that ran parallel to the shore.”

Following this change in the next line, the word “A” was replaced with “Each” to improve readability. 139

143-145 Did the divers count the shells in the field or was all counting down using ImageJ--please clarify. Revision accepted. Rephrased to “In each photo quadrat, the number of oyster shells was counted using the software ImageJ v1.52 by tracing around the shape of the individual shells”

 147

154 Replace “within” with “by” Revision accepted. 158

164 Should MDS be nMDS? Although “MDS” is often used in a general sense to include non-metric MDS, we accept that “nMDS” is technically more precise for our analyses. Revision accepted (also at line 188). 171

172-173 Rephrase to - (“diminishing rate of increase as reef approaches ‘recovered’ condition”) Revision accepted. Rephrased to “(diminishing rate of increase as reef approaches ‘recovered’ condition).” 179

173 The authors should explain why year was log-transformed. Log-transformation of year was used to reflect the non-linear nature of the recovery process, representing it as a logarithmic function of time. Sentence revised to “The response variable was modelled as a function of the log of years since disturbance to represent this non-linear (logarithmic) process.” 181

177-178 Should be in methods Revision accepted line moved to 152. “Video footage was taken of the seafloor covering a 0.5m swath on each side of the transect, representing a total of approximately six hours of footage.”

Remaining line rephrased to “A total of 27 macrofaunal species were recorded in the course of the surveys (S2 Table & S3 Table).” 152

181 Replace “a very good” with “is an accurate representation” Revision accepted 189

192 Table 1 -- Referring now to the treatments and treatment sites is confusing to the reader, I suggest maintaining the naming system above and referring to Treatments 1 and 3 (removing “site”). Revision accepted 200

195-197 It’d be great if in the methods, the authors wrote a quick few sentences about the strengths and weaknesses of Shannon-Wieners, Margalef’s and Pielou’s evenness so the reader can better make sense of the results and what each of these values means. Revision accepted. Rephrased to “including calculation of the biodiversity indices: Margalef’s richness (d) a measure based on the number of species present and sampling effort; Pielou’s evenness (J) a measure of how evenly species are distributed; and Shannon-Wiener’s diversity (H’) which takes both richness and evenness into account.” 165-167

198 Average oyster shell density increased between what treatments? Only 1 and 3? Need to specify. Revision accepted. Rephrased “increased between treatments one and two by 749.5%” 204

217 Rephrase--“…is expected to represent community-wide patterns” Revision accepted. Rephrased “the sampling method presented here is expected to represent community-wide patterns.” 223

218 This is great, I think it’s really important to acknowledge the data that wasn’t collected for this study and could have (but are unlikely to have) influenced results. Instead of “included in the study” replace with “were not evaluated”. Revision accepted. Rephrased “Potential confounding environmental factors were not evaluated” 224

230;233 Replace bottom-contacting with bottom towed. Unless the authors are referring to other fishing practices (shrimp pots, lobster traps, bottom trawling)--if this is the case, some examples in parentheses would be helpful. Revision accepted 237;240

246 Rephrase to “…that trophic cascades could ultimately transform the benthic community and affect oyster reef recovery” Revision accepted. Rephrased to “A six year timeline in the present study is too short to exclude the possibility that trophic cascades ultimately transform the benthic community and affect oyster reef biodiversity.” 256

247-249 I suggest the authors improve the text in this sentence here as it’s unclear what they are trying to say is the reasoning behind why their study is not evidence of intermediate disturbance hypothesis. Also, intermediate disturbance hypothesis appears out of nowhere so perhaps the authors could add a sentence before the current one that addresses other potential explanations to support their findings and then follow it up with why it doesn’t apply (as they’ve done, but with slight editing to make clearer). On reflection we believe this point is far too speculative and have remove the sentence. The paragraph therefore concludes on the far more persuasive consideration of trophic cascade. 256-257

250 What kind of relationship? Positive? Negative? This type of information is important for guiding the reader. Revision accepted. Rephrased to “Logarithmic regression showed positively correlated relationships between biodiversity, oyster shell density and time since disturbance.” 258

250-258 The authors should indicate the directionality of all of the relationships in this paragraph to help guide the reader. Revision accepted. Rephrased to “Logarithmic regression showed positively correlated relationships between biodiversity, oyster shell density and time since disturbance. Oyster shell density was the best statistical predictor of Shannon-Wiener’s diversity, which is likely to be due to the positive relationship between shell density and structural complexity [11][13]. Lown et al. (2021) demonstrated a positive relationship between oyster density and biodiversity using dredge sampling in a remnant oyster fishery in England [11]. The present findings are consistent with Lown et al. (2021), confirming a strong relationship between biodiversity and shell density [11].” 258-264

260-263 Great! 

278 Rephrase slightly: “perhaps the most direct evidence yet quantifying the potential biodiversity gain…” Revision accepted. Rephrased to “The relationship between oyster shell and biodiversity presented here is perhaps the most direct evidence yet quantifying the potential biodiversity gain” 286

280 Replace “would probably double in a decade” to “would likely double within a decade” Revision accepted. Rephrased to 289

281 Make more concise; “the present findings demonstrate the benefits of restoration and the trajectory of restoration success as it relates to biodiversity.” Then, the authors could start a new sentence “Our data showed a link between increased shell material and increased biodiversity. Revision accepted. Rephrased to “Overall, the present findings demonstrate the benefits of restoration and the trajectory of restoration success in terms of biodiversity [1][35][36]. Our data showed a link between increased shell material and increased biodiversity.” 290

283 Missing a period. Revision accepted. 293

284 Rephrase; “rare example from a long-term dredge fishery whose practices appear to have allowed rare oyster habitat and the associated community to persist, thereby providing a valuable insight into the recovery of biodiversity by balancing sustainable fishery practices with European oyster habitat recovery efforts”. Since this study did not track restoration over time but recovery post disturbance, distinguishing between the two is critical. Revision accepted. Rephrased to “The present study is a rare example from a long-term dredge fishery [40][48] whose practices appear to have allowed rare oyster habitat and the associated community to persist, thereby providing a valuable insight into the recovery of biodiversity by balancing sustainable fishery practices with European oyster habitat recovery efforts.”

 294

The following change was also made:

Figure caption 1. “Lefnol point” was corrected to “Lefnol Point”

We look forward to hearing from you in due time regarding our submission and to respond to

any further questions and comments you may have.

Sincerely,

Naomi Kennon

24/01/23

---

## [Editor Report · Decision Letter 4]

7 Mar 2023

Rotational fishing enables biodiversity recovery and provides a model for oyster (Ostrea edulis) habitat restoration

PONE-D-22-14818R4

Dear Dr. Kennon,

We’re pleased to inform you that your manuscript has been judged scientifically suitable for publication and will be formally accepted for publication once it meets all outstanding technical requirements.

Kind regards,

José A. Fernández Robledo, Ph.D.

Academic Editor

PLOS ONE
---

## [Editor Report · Acceptance letter]

13 Mar 2023

PONE-D-22-14818R4 

­­Rotational fishing enables biodiversity recovery and provides a model for oyster (*Ostrea edulis*) habitat restoration 

Dear Dr. Kennon:

I'm pleased to inform you that your manuscript has been deemed suitable for publication in PLOS ONE. Congratulations! Your manuscript is now with our production department. 

Kind regards, 

on behalf of

Dr. José A. Fernández Robledo 

Academic Editor

PLOS ONE